# IMPACT OF AGENT BEHAVIOR IN DISTRIBUTED SGD AND FEDERATED LEARNING

## ABSTRACT

Distributed learning has gained significant interest recently as it allows for the training of machine learning models across a set of *heterogeneous* agents in a privacy-preserving manner with the growing amount of distributed data. In this paper, we conduct an asymptotic analysis of Generalized Distributed SGD (GD-SGD) under various communication patterns among agents, including Distributed SGD (D-SGD) and its variants in Federated Learning (FL), as well as the increasing communication interval in the FL setting. We examine the influence of agents' sampling strategies, such as *i.i.d.* sampling, shuffling methods and Markovian sampling, on the overall convergence speed of GD-SGD. We prove that all agents will asymptotically reach consensus and identify the optimal model parameter, while also analyzing the impact of sampling strategies on the limiting covariance matrix that appears in the Central Limit Theorem (CLT). Our results theoretically and empirically support recent findings on linear speedup and asymptotic network independence, and generalize previous findings on the efficient Markovian sampling strategies from vanilla SGD to GD-SGD. Overall, our results provide a deeper understanding of the convergence speed of GD-SGD and emphasize the role of *each* agent's sampling strategy, moving beyond a focus on the worst-case agent commonly found in existing literature.

## 1 INTRODUCTION

Distributed learning deals with the training of models across multiple agents over a communication network in a distributed manner, while addressing the challenges of privacy, scalability, and high-dimensional data (Boyd et al., 2011; McMahan et al., 2017). Each agent $i \in [N]$ holds a private dataset $\mathcal{X}_i$ and an agent-specified loss function $F_i : \mathbb{R}^d \times \mathcal{X}_i \to \mathbb{R}$ that depends on the model parameter $\theta \in \mathbb{R}^d$ and a data point $X \in \mathcal{X}_i$. The goal is then to find

$$\theta^* \in \mathcal{L} \triangleq \underset{\theta \in \mathbb{R}^d}{\arg\min} \left\{ f(\theta) \triangleq \frac{1}{N} \sum_{i=1}^N f_i(\theta) \right\}, \tag{1}$$

where $\mathcal{L}$ is the set of minimizers, the local function $f_i(\theta) \triangleq \mathbb{E}_{X \sim \mathcal{D}_i}[F_i(\theta, X)]$ and $\mathcal{D}_i$ represents the target distribution of data for agent $i$.[1] Due to the distributed nature, $\{\mathcal{D}_i\}_{i \in [N]}$ and $\{\mathcal{X}_i\}_{i \in [N]}$ are not necessarily identically distributed over $[N]$. We assume each agent $i$ can locally compute the gradient $\nabla F_i(\theta, X) \in \mathbb{R}^d$ w.r.t. $\theta$ for every $X \in \mathcal{X}_i$. To solve the optimization problem (1), we consider the Generalized Distributed SGD (GD-SGD), as outlined in Koloskova et al. (2020), which encompasses Distributed SGD (D-SGD) algorithm (Wai, 2020; Wang et al., 2020a; Olshevsky, 2022; Sun et al., 2023), as well as its variants in Federated Learning (FL) (McMahan et al., 2017; Woodworth et al., 2020; Li et al., 2022). At time $n$, each agent $i \in [N]$ updates its model parameter $\theta_{n+1}^i$ as follows:

$$\text{Local update: } \theta_{n+1/2}^i = \theta_n^i - \gamma_{n+1} \nabla F_i(\theta_n^i, X_n^i), \tag{2a}$$

$$\text{Aggregation: } \theta_{n+1}^i = \sum_{j=1}^N w_n(i,j) \theta_{n+1/2}^j, \tag{2b}$$

where $\gamma_n$ denotes the step size, $X_n^i$ is the data point sampled by agent $i$ at time $n$, and $\mathbf{W}_n = [w_n(i,j)]_{i,j \in [N]}$ represents the doubly-stochastic communication matrix. Note that (2) reduces to the vanilla SGD when $N = 1$ (and thus $\mathbf{W}_n = 1$ for all time $n$).

---

[1]Throughout the paper we don't impose convexity assumption on the objective function $f(\theta)$. For non-convex function $f(\theta)$, the target is to find a critical point $\theta^*$ for which $\nabla f(\theta^*) = \mathbf{0}$.

**Versatile Communication Patterns $\{\mathbf{W}_n\}$:** For visualization, we depict the scenarios of the GD-SGD algorithm (2) in Figure 1. Specifically, in the D-SGD algorithm, each agent, represented by a node in the graph, communicates with its neighbors after each SGD computation via $\mathbf{W}_n$, representing the underlying network topology. A central server-based aggregation can also be employed, leading to a fully connected network among agents, with $\mathbf{W}_n$ degenerating to a rank-1 matrix $\mathbf{W}_n = \mathbf{1}\mathbf{1}^T/N$. To minimize communication expenses, FL variants allow each agent to perform multiple SGD steps before aggregating with their neighbors or a central server (McMahan et al., 2017; Stich, 2018; Woodworth et al., 2020). As a result, FL

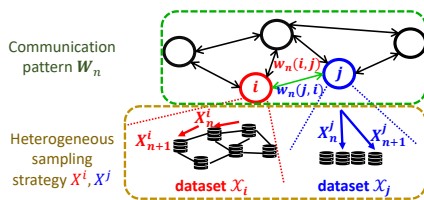

Figure 1: GD-SGD algorithm with a communication network of $N = 5$ agents, each holding potentially distinct datasets; e.g., agent $j$ (in blue) samples $\mathcal{X}_j$ *i.i.d.* and agent $i$ (in red) samples $\mathcal{X}_i$ via Markovian trajectory.

variants feature a communication interval of length $K$ and a communication pattern $\mathbf{W}_n = \mathbf{W}$ for $n = mK, \forall m \in \mathbb{N}$ and $\mathbf{W}_n = \mathbf{I}_N$ otherwise in (2).[2] In particular, i) $\mathbf{W} = \mathbf{1}\mathbf{1}^T/N$ corresponds to local SGD with full client participation (L-SGD-FC), where all clients take part in each round (Khodadadian et al., 2022; Woodworth et al., 2020; Li et al., 2022); ii) $\mathbf{W}$ is a random matrix generated by partial client sampling (L-SGD-PC), where only a random subset of clients participate in each round (McMahan et al., 2017; Chen et al., 2022; Wang & Ji, 2022); iii) $\mathbf{W}$ can be generated by Metropolis-Hasting algorithm concerning the underlying network topology in hybrid local SGD (HL-SGD) (Hosseinalipour et al., 2022; Guo et al., 2022) and decentralized FL (D-FL) (Lalitha et al., 2018; Ye et al., 2022; Chellapandi et al., 2023). We will provide a comprehensive discussion on incorporating these types of matrices $\mathbf{W}$ into the GD-SGD algorithm in Appendix B.

**Markovian vs *i.i.d.* Sampling:** Agents typically employ *i.i.d.* or Markovian sampling, as illustrated in the bottom brown box of Figure 1. In cases where agents have full access to their data, D-SGD with *i.i.d* sampling has been extensively studied from both asymptotic (Mathkar & Borkar, 2016; Morral et al., 2017) and non-asymptotic (Neglia et al., 2020; Koloskova et al., 2020; Olshevsky, 2022; Le Bars et al., 2023) perspectives. In the FL literature, Li et al. (2022) specifically addressed the Central Limit Theorem (CLT) for L-SGD-FC with increasing communication intervals, while Chen et al. (2022); Ye et al. (2022); Guo et al. (2022); Hosseinalipour et al. (2022); Liao et al. (2023) investigated other application-oriented FL variants to deal with the communication costs. In addition, a unified framework for GD-SGD was analyzed in Koloskova et al. (2020). However, all of these works solely focus on *i.i.d.* sampling, restricting their applicability to Markovian sampling scenarios.

The study of Markovian sampling under limited settings (shown in Table 1) has recently received increased attention, where agents may not have independent access to their data. For instance, in statistical applications, agents may have an unknown a priori distribution of the dataset and it is common to use Markovian sampling instead of *i.i.d.* sampling (Jerrum & Sinclair, 1996; Robert et al., 1999). In the context of HL-SGD with multiple device-to-device (D2D) networks (Guo et al., 2022; Hosseinalipour et al., 2022), using a random walk over each D2D network can reduce communication costs (Hu et al., 2022; Even, 2023; Ayache et al., 2023), as opposed to the Gossip algorithm in Guo et al. (2022) that necessitates frequent parameter aggregation. The special case of a single agent employing a random walk over a D2D network, as an application of vanilla SGD with Markovian noise, has been widely studied in the literature with the improved communication efficiency and privacy guarantees (Sun et al., 2018; Hu et al., 2022; Triastcyn et al., 2022; Even, 2023; Hendrikx, 2023). On the other hand, agents with full access to their datasets can adopt more efficient methods than *i.i.d.* sampling. In high-dimensional and combinatorial spaces with constraints, *i.i.d.* sampling using the acceptance-rejection method Brémaud (2013) can be computationally expensive due to multiple rejections before obtaining a sample that satisfies the constraints, leading to wasted samples (Duchi et al., 2012; Sun et al., 2018). Alternatively, agents can use more sampling-efficient Markov Chain Monte Carlo (MCMC) methods, such as those in Dyer et al. (1993); Jerrum & Sinclair (1996). In addition, shuffling methods can be considered as high-order Markov chains (Hu et al., 2022), which achieves faster convergence than *i.i.d.* sampling (Ahn et al., 2020; Yun et al., 2021; 2022).

**Influence of Agent's Sampling Strategy:** Some recent works have studied the non-asymptotic behavior of D-SGD and FL variants under Markovian sampling, as illustrated in Table 1. However,

---

[2]In this paper, we extend Li et al. (2022) from *i.i.d.* sampling to Markovian sampling and additional communication patterns in this paper, allowing the length $K$ to increase gradually. This slows the aggregation frequency and further reduces communication costs while maintaining convergence to the optimal point $\theta^*$.

Table 1: Recent works in the D-SGD and FL literature: We classify the communication patterns into five categories, i.e., D-SGD, L-SGD-FC, L-SGD-PC, HL-SGD and D-FL. We mark 'GD-SGD' when all five aforementioned patterns are included. Abbreviations: '**Asym.**' = 'Asymptotic', '**Comm.**' = 'Communication', '**D.A.B**' = 'Differentiating Agent Behavior', '**L.S.**' = 'Linear Speedup', '**A.N.I.**' = 'Asymptotic Network Independence', '**I.C.I.**' = 'Increasing Communication Interval $K$'.

| Reference | Analysis | Sampling | Comm. Pattern | D.A.B. | L.S. | A.N.I. | I.C.I. |
|---|---|---|---|---|---|---|---|
| Morral et al. (2017) | Asym. | *i.i.d.* | D-SGD | ✓ | ✓ | ✓ | N/A |
| Li et al. (2022) | Asym. | *i.i.d.* | L-SGD-FC | ✓ | ✓ | N/A | ✓ |
| Koloskova et al. (2020); Le Bars et al. (2023) | Non-Asym. | *i.i.d.* | GD-SGD | × | ✓ | × | × |
| Olshevsky (2022) | Non-Asym. | *i.i.d.* | D-SGD | × | × | ✓ | N/A |
| Chen et al. (2022); Liao et al. (2023) | Non-Asym. | *i.i.d.* | L-SGD-PC | × | ✓ | N/A | × |
| Hosseinalipour et al. (2022); Guo et al. (2022) | Non-Asym. | *i.i.d.* | HL-SGD | × | ✓ | × | N/A |
| Ye et al. (2022); Chellapandi et al. (2023) | Non-Asym. | *i.i.d.* | D-FL | × | ✓ | × | × |
| Wai (2020); Zeng et al. (2022); Sun et al. (2023) | Non-Asym. | Markov | D-SGD | × | × | × | N/A |
| Khodadadian et al. (2022); Wang et al. (2023) | Non-Asym. | Markov | L-SGD-FC | × | ✓ | N/A | × |
| Doan et al. (2019) | Non-Asym. | Markov | GD-SGD | × | ✓ | × | × |
| Ayache et al. (2023); Even (2023) | Non-Asym. | Markov | N/A | N/A | N/A | N/A | N/A |
| Hu et al. (2022); Li et al. (2023) | Asym. | Markov | N/A | N/A | N/A | N/A | N/A |
| Our Work | Asym. | Markov | GD-SGD | ✓ | ✓ | ✓ | ✓ |

the finite-time upper bounds in these works are not sharp enough to unveil the real statistical information on *each* agent's sampling strategy in GD-SGD. Specifically, Wai (2020); Sun et al. (2023) proposed the error bound $O(\frac{1/\log^2(1/\rho)}{n^{1-a}})$, where $a \in (0.5, 1]$ and $\rho$ is the mixing rate of the underlying Markov chain for each agent, which is assumed to be identical for all agents, ignoring the agent heterogeneity. Similar assumption was also made in Khodadadian et al. (2022). Recently, Zeng et al. (2022); Wang et al. (2023) relaxed this assumption but they only considered the finite-time bound of the form $O(\tau_{mix}^2/(n+1))$, where $\tau_{mix}$ represents the mixing time corresponding to the Markov chain that mixes the slowest and is interpreted as *the worst-performing agent*.[3] In other words, the bounds in these works would remain the same even when some other agents employ better Markov chains with faster mixing rates. Therefore, they are unable to capture the effect of *other* agents on the overall system performance. This is a significant shortcoming, particularly in large-scale machine learning applications where *the* worst-performing agent may be difficult to find and control (due to strong privacy concern or sporadic unreachability). Since agents in distributed learning can decide how to sample their local datasets, it is thus essential to understand how improvements in each agent's sampling strategy translate into the overall convergence speed of the GD-SGD algorithm.

**Rationale for Asymptotic Analysis:** Although non-asymptotic analysis has been favored over asymptotic analysis in recent years, it is worth noting that *asymptotic and non-asymptotic analyses are both important and informative, and their combination provides a more comprehensive understanding of the convergence behavior*, as indicated in Borkar et al. (2021); Orabona (2020); Mou et al. (2020); Li & Milzarek (2022). For vanilla SGD, Mou et al. (2020); Chen et al. (2020) emphasized that CLT is far less asymptotic than it may appear under both *i.i.d.* and Markovian sampling. Specifically, the limiting covariance matrix seen in the CLT, which is the statistical information of vanilla SGD, also appears in the high-probability bound (Mou et al., 2020) and the explicit finite-time bound (Chen et al., 2020). Additionally, Hu et al. (2022) numerically showed that the limiting covariance matrix can capture the convergence more accurately than the mixing rate commonly used in finite-time upper bounds in Duchi et al. (2012); Sun et al. (2018). Furthermore, Hu et al. (2022) argued that finite-time analysis is unsuitable for some efficient high-order Markov chains because the mixing-rate-based comparisons between the high-order Markov chains and their baselines are unavailable. However, a similar comparison of each agent's sampling strategy and its impact on overall performance is still missing when extending from vanilla SGD to GD-SGD.

**Our Contributions:** In this paper, we present an asymptotic analysis of the GD-SGD algorithm (2) under heterogeneous Markovian sampling $\{X_n^i\}$ and a large family of communication patterns $\{\mathbf{W}_n\}$ among agents, including D-SGD and its FL variants, as well as the increasing communication interval in the FL setting shown in Table 1. This enables us to differentiate the contribution of each agent in the learning process and to gain insights into the impact of their sampling strategies on the overall convergence speed of the GD-SGD algorithm. Main results are summarized as follows:

---

[3]While improving the finite-time upper bound to distinguish each agent may not be the focus of the aforementioned works, their analyses require every Markov chain to be close to some neighborhood of its stationary distribution. This naturally incurs a maximum operator, and thus convergence is strongly influenced by the slowest mixing rate, i.e., the worst-performing agent.

- We show that under suitable assumptions, all agents performing (2) will asymptotically reach the consensus and find the optimal model parameter $\theta^*$, i.e.,

$$\lim_{n \to \infty} \|\theta_n^i - \theta_n\| = 0, \forall i \in [N], \quad \lim_{n \to \infty} \|\theta_n - \theta^*\| = 0 \text{ a.s.} \tag{3}$$

where $\theta_n \triangleq \frac{1}{N} \sum_{i=1}^{N} \theta_n^i$ is the average model parameter. Furthermore, define $\bar{\theta}_n \triangleq \frac{1}{n} \sum_{s=0}^{n-1} \theta_s$, we derive the CLT in the form of

$$\sqrt{n}(\bar{\theta}_n - \theta^*) \xrightarrow[n \to \infty]{dist.} \mathcal{N}\left(0, \bar{\mathbf{V}}'/N\right) \tag{4}$$

where $\bar{\mathbf{V}}' \triangleq \frac{1}{N} \sum_{i=1}^{N} \mathbf{V}_i'$ and $\mathbf{V}_i'$ is the limiting covariance matrix of agent $i$, which depends mainly on the sampling strategy $\{X_n^i\}$ of agent $i$.

- We provide a comprehensive understanding of the collective impact of the sampling strategies employed by all agents on average limiting covariance matrix $\bar{\mathbf{V}}'$. This differs from the previous non-asymptotic analysis, which only revealed the effect of the worst-performing agent on the finite-time bounds. Our CLT result illustrates that even when the worst-performing agent can't be improved, $\bar{\mathbf{V}}'$ can still be reduced, and such a reduction can be interpreted as the overall acceleration of the convergence with smaller mean-square error (MSE). This can be achieved by refining the sampling strategies of other agents using established efficient methods, i.e., utilizing shuffling methods in place of *i.i.d.* sampling (Bottou, 2012; Ahn et al., 2020; Yun et al., 2021; 2022), or even incorporating non-Markovian processes studied in the MCMC literature (Lee et al., 2012; Li et al., 2015; 2019).

- We demonstrate that our analysis supports recent findings from studies such as Khodadadian et al. (2022); Wang et al. (2023), which exhibited linear speedup scaling with the number of agents under L-SGD-FC with Markovian sampling; and Pu et al. (2020); Olshevsky (2022), which examined the notion of 'asymptotic network independence' for D-SGD with *i.i.d.* sampling, where the convergence of the algorithm (2) at large time $n$ depends solely on the left eigenvector of $\mathbf{W}_n$ ($\mathbf{1}/N$ considered in this paper) rather than the specific communication network topology encoded in $\mathbf{W}_n$, but all now under Markovian sampling. We extend these findings in view of CLT to a broader range of communication patterns $\{\mathbf{W}_n\}$ and Markovian sampling strategies $\{X_n^i\}$.

- Our setting and results are general enough in that they reduce to recent findings in Hu et al. (2022) as a special case with $N = 1$. This implies that all the results in Hu et al. (2022), such as the relationship between the efficiency of the Markov chain and the limiting covariance matrix in the CLT of vanilla SGD, can carry over to our GD-SGD setting. In contrast to finite-time bounds that only capture the worst-case scenario out of $N$ agents, our CLT result (4) directly reflects the contribution of each agent's sampling strategy to the overall convergence speed of GD-SGD.

- We present simulations to demonstrate the impact of agents' sampling strategies, the length of communication interval in the FL setting, and communication patterns on the overall convergence. Our results suggest that utilizing efficient sampling strategies for some agents, as determined through asymptotic analysis, leads to a reduction in the MSE over the majority of the time periods.

## 2 PRELIMINARIES

**Basic Notations:** We use $\|\mathbf{v}\|$ to indicate the Euclidean norm of a vector $\mathbf{v} \in \mathbb{R}^d$ and $\|\mathbf{M}\|$ to indicate the spectral norm of a matrix $\mathbf{M} \in \mathbb{R}^{d \times d}$. The identity matrix of dimension $d$ is denoted by $\mathbf{I}_d$, and the all-one (resp. all-zero) vector of dimension $N$ is denoted by $\mathbf{1}$ (resp. $\mathbf{0}$). Let $\mathbf{J} \triangleq \mathbf{1}\mathbf{1}^T/N$. The diagonal matrix with the entries of $\mathbf{v}$ on the main diagonal is written as $\text{diag}(\mathbf{v})$. We also use '$\succeq$' for Loewner ordering such that $\mathbf{A} \succeq \mathbf{B}$ is equivalent to $\mathbf{x}^T(\mathbf{A} - \mathbf{B})\mathbf{x} \geq 0$ for any $\mathbf{x} \in \mathbb{R}^d$.

**Asymptotic Covariance Matrix:** Asymptotic variance is a widely used metric for evaluating the second-order properties of Markov chains associated with a scalar-valued test function in the MCMC literature, e.g., Chapter 6.3 Brémaud (2013), and asymptotic covariance matrix is its multivariate version for a vector-valued function. Specifically, we consider a finite, irreducible, aperiodic and positive recurrent (ergodic) Markov chain $\{X_n\}_{n \geq 0}$ with transition matrix $\mathbf{P}$ and stationary distribution $\boldsymbol{\pi}$, and the estimator $\hat{\mu}_n(\mathbf{g}) \triangleq \frac{1}{n} \sum_{s=0}^{n-1} \mathbf{g}(X_s)$ for any vector-valued function $\mathbf{g} : [N] \to \mathbb{R}^d$. According to the ergodic theorem Brémaud (2013); Brooks et al. (2011), we have $\lim_{n \to \infty} \hat{\mu}_n(\mathbf{g}) = \mathbb{E}_{\boldsymbol{\pi}}(\mathbf{g})$ a.s.. As defined in Brooks et al. (2011); Hu et al. (2022), the asymptotic covariance matrix $\boldsymbol{\Sigma}_X(\mathbf{g})$ for a vector-valued function $\mathbf{g}(\cdot)$ is given by

$$\boldsymbol{\Sigma}_X(\mathbf{g}) \triangleq \lim_{n \to \infty} n \cdot \text{Var}(\hat{\mu}_n(\mathbf{g})) = \lim_{n \to \infty} \mathbb{E}\left\{\Delta_n \Delta_n^T\right\}/n, \tag{5}$$

where $\Delta_n \triangleq \sum_{s=0}^{n-1}(\mathbf{g}(X_s) - \mathbb{E}_{\boldsymbol{\pi}}(\mathbf{g}))$. By following the algebraic manipulations in Theorem 6.3.7 of Brémaud (2013) for asymptotic variance (univariate version), we can rewrite (5) in a matrix form

$$\boldsymbol{\Sigma}_X(\mathbf{g}) = \mathbf{G}^T \text{diag}(\boldsymbol{\pi}) \left( \mathbf{Z} - \mathbf{I}_N + \mathbf{1}\boldsymbol{\pi}^T \right) \mathbf{G}, \tag{6}$$

where $\mathbf{G} \triangleq [\mathbf{g}(1), \cdots, \mathbf{g}(N)]^T \in \mathbb{R}^{N \times d}$ and $\mathbf{Z} \triangleq [\mathbf{I}_N - \mathbf{P} + \mathbf{1}\boldsymbol{\pi}^T]^{-1}$. This matrix form explicitly shows the dependence on the transition matrix $\mathbf{P}$ and its stationary distribution $\boldsymbol{\pi}$.

**Model Description:** The GD-SGD in (2) can be expressed in a compact iterative form, i.e.,

$$\theta_{n+1}^i = \sum_{j=1}^N w_n(i,j)(\theta_n^j - \gamma_{n+1}\nabla F_j(\theta_n^j, X_n^j)), \tag{7}$$

where each agent $i$ samples according to its own Markovian trajectory $\{X_n^i\}_{n \geq 0}$ with stationary distribution $\pi_i$ such that $\mathbb{E}_{X \sim \pi_i}[F_i(\theta, X)] = f_i(\theta)$. Let $K_l$ denote the communication interval between the $(l-1)$-th and $l$-th aggregation among $N$ agents, and let $n_l \triangleq \sum_{m=1}^l K_m$ be the time instance of the $l$-th aggregation. Also, let $\tau_n \triangleq \min_l\{l : n_l \geq n\}$ denote the next aggregation at time $n$. Thus, $K_{\tau_n}$ corresponds to the communication interval for the $\tau_n$-th aggregation, which includes time index $n$. If $n \neq n_l$, agents perform individual SGD iterations, so $\mathbf{W}_n = \mathbf{I}_N$; otherwise, $\mathbf{W}_n = \mathbf{W}$ for aggregation between agents. We note three things: i) for $K_l = 1$, (7) simplifies to D-SGD; ii) for $K_l = K > 1$, (7) corresponds to FL variants. iii) When $K_l$ increases with $l$, we recover some choices of $K_l$ previously explored in Li et al. (2022) for L-SGD-FC with *i.i.d.* sampling. This increasing communication interval aims to further reduce the frequency of aggregation among agents for lower communication costs, but now under a Markovian sampling setting and a wider range of communication patterns. We below state the needed assumptions.

**Assumption 2.1** (Regularity of the gradient). *For each $i \in [N]$ and $X \in \mathcal{X}^i$, the function $F_i(\theta, X)$ is $L$-smooth in terms of $\theta$, i.e., for any $\theta_1, \theta_2 \in \mathbb{R}^d$,*

$$\|\nabla F_i(\theta_1, X) - \nabla F_i(\theta_2, X)\| \leq L\|\theta_1 - \theta_2\|. \tag{8}$$

*In addition, $f$ in (1) is locally strongly convex around the minimizer $\theta^* \in \mathcal{L}$, i.e.,*

$$\mathbf{H} \triangleq \nabla^2 f(\theta^*) \succeq \mu \mathbf{I}_d. \tag{9}$$

Assumption 2.1 imposes the regularity conditions on the gradient $\nabla F_i(\cdot, X)$ and Hessian matrix of the objective function $f(\cdot)$, as is commonly assumed in Borkar (2009); Kushner & Yin (2003); Fort (2015); Hu et al. (2022). Note that (8) requires per-sample Lipschitzness of $\nabla F_i$ and is stronger than the Lipschitzness of its expected version $\nabla f_i$, which is commonly assumed under *i.i.d* sampling setting, e.g., Wang et al. (2020b); Li et al. (2020); Fraboni et al. (2022). However, we remark that this is in line with the prior works on D-SGD and L-SGD-FC under Markovian sampling as well, e.g., Wai (2020); Khodadadian et al. (2022); Zeng et al. (2022), because $\nabla F_i(\theta, X)$ is no longer the unbiased stochastic version of $\nabla f_i(\theta)$ and the effect of $\{X_n^i\}$ has to be taken into account in the analysis. The local strong convexity at the minimizer is commonly assumed to analyze the convergence of the algorithm under both asymptotic and non-asymptotic analysis, e.g., Borkar (2009); Fort (2015); Hu et al. (2022); Kushner & Yin (2003); Li et al. (2023); Zeng et al. (2022). Moreover, Appendix C in Hu et al. (2022) showed that (9) does not impose the convexity on $f(\cdot)$ and is no stricter than the widely used Polyak-Lojasiewicz condition (Ahn et al., 2020; Wojtowytsch, 2021; Yun et al., 2022).

**Assumption 2.2** (Ergodicity of Markovian sampling). *$\{X_n^i\}_{n \geq 0}$ is an ergodic Markov chain with stationary distribution $\pi_i$, e.g., $\mathbb{E}_{X \sim \pi_i}[F_i(\theta, X)] = f_i(\theta)$, and is mutually independent over $i \in [N]$.*

The ergodicity of the underlying Markov chains, as stated in Assumption 2.2, is commonly assumed in the literature (Duchi et al., 2012; Sun et al., 2018; Zeng et al., 2022; Khodadadian et al., 2022; Hu et al., 2022). This assumption ensures the asymptotic unbiasedness of the loss function $F_i(\theta, \cdot)$, which takes *i.i.d.* sampling as a special case.

**Assumption 2.3** (Decreasing step size and slowly increasing communication interval). *For communication interval $K_{\tau_n}$, i) if $K_{\tau_n} \leq K$ for all $n$, we assume the polynomial step size $\gamma_n = 1/n^a$ and $a \in (0.5, 1]$; ii) if $K_{\tau_n} \to \infty$ as $n \to \infty$, we assume $\gamma_n = 1/n$ and define $\eta_n = \gamma_n K_{\tau_n}^{L+1}$, where the sequence $\{K_l\}_{l \geq 0}$ satisfies $\sum_n \eta_n^2 < \infty$, $K_{\tau_n} = o(\gamma_n^{-1/2(L+1)})$, and $\lim_{l \to \infty} \eta_{n_l+1}/\eta_{n_{l+1}+1} = 1$.*

In Assumption 2.3-i), the polynomial step size $\gamma_n$ is standard in the literature and it has the property $\sum_n \gamma_n = \infty$, $\sum_n \gamma_n^2 < \infty$ (Chen et al., 2020; Li et al., 2022; Hu et al., 2022). Inspired by Li et al. (2022), we introduce $\eta_n$ to control the step size within each $l$-th communication interval with length $K_l$ to restrict the growth of $K_l$. Specifically, $\sum_n \eta_n^2 < \infty$ and $K_{\tau_n} = o(\gamma_n^{-1/2(L+1)})$ ensure that $\eta_n \to 0$ and $K_{\tau_n}$ should not increase too fast in $n$. $\lim_{l \to \infty} \eta_{n_l+1}/\eta_{n_{l+1}+1} = 1$ sets the restriction on the increment from $n_l$ to $n_{l+1}$. Several practical forms of $K_l$ suggested by Li et al. (2022) including $K_l \sim \log(l)$ and $K_l \sim \log\log(l)$ also satisfy Assumption 2.3-ii). We defer to Appendix A the mathematical verification of these two types of $K_l$.

*Remark* 1. Assumption 2.3-ii) covers the case of increasing communication intervals with the step size $\gamma_n = 1/n$. The conditions on $K_l$, while slightly more stringent than those in Li et al. (2022), are necessary for our analysis *under Markovian sampling*. Note that Assumption 3.2 in Li et al. (2022) only works under *i.i.d* sampling case. Under Markovian sampling, $\nabla F_i(\theta, X) - \nabla f_i(\theta)$ is not unbiased, nor Martingale difference such that the way Li et al. (2022) used Martingale CLT cannot be extrapolated to the Markovian sampling as is. Instead, we referred to the techniques in Fort (2015); Morral et al. (2017) in our way to cover the increasing communication interval under Markovian sampling, at the price of requiring the step size to be $\gamma_n = 1/n$, which was not included in Li et al. (2022). Relaxing this assumption for more general forms of $K_l$ is beyond the scope of this paper.

**Assumption 2.4** (Stability on model parameter). $\sup_n \|\theta_n^i\| < \infty$ *almost surely for all* $i \in [N]$.

Assumption 2.4 claims that the sequence of $\{\theta_n^i\}$ always remains in a path-dependent compact set. It is to ensure the stability of the algorithm that serves the purpose of analyzing the convergence, which is often assumed under the asymptotic analysis of vanilla SGD with Markovian noise (Delyon et al., 1999; Fort, 2015; Li et al., 2023). It is weaker than the uniformly bounded per-sample gradients $\|\nabla F_i(\theta, X)\| < D$ which is considered a strong assumption in the literature. As mentioned in Morral et al. (2017); Vidyasagar (2022), checking Assumption 2.4 is challenging and requires case-by-case analysis, even under *i.i.d.* sampling. Only recently the stability of SGD under Markovian sampling has been studied in Borkar et al. (2021), but the result for GD-SGD remains unknown in the literature. Thus, we analyze each agent's sampling strategy in the asymptotic regime under this stability condition.

**Assumption 2.5** (Contraction property of communication matrix). *i).* $\{\mathbf{W}_n\}_{n\geq 0}$ *is independent of the sampling strategy* $\{X_n^i\}_{n\geq 0}$ *for all* $i \in [N]$ *and is assumed to be doubly-stochastic for all* $n$; *ii). At each aggregation step* $n_l$, $\bar{\mathbf{W}}_{n_l}$ *is independently generated from some distribution* $\mathcal{P}_{n_l}$ *such that* $\|\mathbb{E}_{\mathbf{W}\sim\mathcal{P}_{n_l}}[\mathbf{W}^T\mathbf{W}] - \mathbf{J}\| \leq C_1 < 1$ *for some constant* $C_1$.

The doubly-stochasticity of $\mathbf{W}_n$ in Assumption 2.5-i) is widely assumed in the literature (Mathkar & Borkar, 2016; Doan et al., 2019; Koloskova et al., 2020; Zeng et al., 2022). Assumption 2.5-ii) is a contraction property to ensure that agents employing GD-SGD will asymptotically achieve the consensus, which is also commonly seen in Bianchi et al. (2013); Doan et al. (2019); Zeng et al. (2022). Examples of $\mathbf{W}$ that satisfy Assumption 2.5-ii), e.g., Metropolis-Hasting matrix, partial client sampling in FL, are deferred to Appendix B due to space constraint.

# 3 ASYMPTOTIC ANALYSIS OF GD-SGD

## 3.1 MAIN RESULTS

**Almost-sure Convergence:** Denote by $\theta_n \triangleq \frac{1}{N}\sum_{i=1}^{N}\theta_n^i$ the consensus among all the agents at time $n$, we establish the asymptotic consensus of the local parameters $\theta_n^i$, as stated in Lemma 3.1.

**Lemma 3.1.** *With Assumptions 2.1, 2.3, 2.4 and 2.5, the GD-SGD iteration* (7) *under Markovian sampling leads to the consensus almost surely, i.e.,*

$$\lim_{n\to\infty} \|\theta_n^i - \theta_n\| = 0, \forall i \in [N] \quad a.s. \tag{10}$$

Lemma 3.1 can be seen as an extension of Proposition 1 in Morral et al. (2017) but now incorporated with Markovian sampling, FL setting, and increasing communication interval $K_l$ (with Assumption 2.3). We provide its proof in Appendix C, while the main difficulty lies in showing the boundedness of a sequence $\{\gamma_n^{-1}(\theta_n^i - \theta_n)\}_{n\geq 0}$ almost surely for all $i \in [N]$, which is proved in Lemma C.1. Next, with additional Assumption 2.2, we are able to obtain the almost-sure convergence to $\theta^* \in \mathcal{L}$.

**Theorem 3.2.** *Under Assumptions 2.1 - 2.5, the consensus* $\theta_n$ *converges to* $\mathcal{L}$ *almost surely, i.e.,*

$$\limsup_n \inf_{\theta^*\in\mathcal{L}} \|\theta_n - \theta^*\| = 0 \quad a.s. \tag{11}$$

Theorem 3.2 follows from the convergence guarantees in Theorem 2 of Delyon et al. (1999) by decomposing the Markovian noise term $\nabla F_i(\theta_n^i, X_n^i) - \nabla f_i(\theta_n^i)$ by the Poisson equation technique (Benveniste et al., 2012; Fort, 2015; Chen et al., 2020) into a Martingale difference noise term, along with additional noise terms, and transforming (7) into a stochastic-approximation-like iteration (which is shown in (57)) and verifying all the conditions on the noise terms therein under the given assumptions here. It also implies that GD-SGD guarantees almost-sure convergence to an optimal

point $\theta^* \in \mathcal{L}$ for every agent, even when $K_l$ increases in $l$, as allowed by Assumption 2.3-ii). The proof is deferred to Appendix D.

**Central Limit Theorem:** Denote by $\mathbf{U}_i \triangleq \mathbf{\Sigma}_{X^i}(\nabla F_i(\theta^*, \cdot))$ the asymptotic covariance matrix (defined in (6)) associated with agent $i \in [N]$, given its sampling strategy $\{X_n^i\}$ and function $\nabla F_i(\theta^*, \cdot)$. Let $\mathbf{U} \triangleq \frac{1}{N^2} \sum_{i=1}^{N} \mathbf{U}_i$. We assume the polynomial step-size $\gamma_n \sim \gamma_\star / n^a$, $a \in (0.5, 1]$ and $\gamma_\star > 0$. In the case of $a = 1$, we further assume $\gamma_\star > 1/2\mu$, where $\mu$ is defined in (9).

**Theorem 3.3.** *Let Assumptions 2.1 - 2.5 hold. Then,*

$$\gamma_n^{-1/2}(\theta_n - \theta^*) \xrightarrow[n\to\infty]{dist.} \mathcal{N}(0, \mathbf{V}), \tag{12}$$

*where the limiting covariance matrix $\mathbf{V}$ is in the form of*

$$\mathbf{V} = \int_0^\infty e^{\mathbf{M}t} \mathbf{U} e^{\mathbf{M}^T t} dt. \tag{13}$$

*Here, we have $\mathbf{M} = -\mathbf{H}$ if $a \in (0.5, 1)$, or $\mathbf{M} = \mathbf{I}_d / 2\gamma_\star - \mathbf{H}$ if $a = 1$, where $\mathbf{H}$ is defined in (9).*

*Moreover, let $\bar{\theta}_n = \frac{1}{n} \sum_{s=0}^{n-1} \theta_s$. Then, for $a \in (0.5, 1)$ and let $\mathbf{V}' \triangleq \mathbf{H}^{-1} \mathbf{U} \mathbf{H}^{-T}$, we have*

$$\sqrt{n}(\bar{\theta}_n - \theta^*) \xrightarrow[n\to\infty]{dist.} \mathcal{N}(0, \mathbf{V}'). \tag{14}$$

The proof of Theorem 3.3 is included in Appendix E. To obtain the CLT result, we need to quantify the second-order conditions for the decomposition of the Markovian noise term that are absent in the *i.i.d.* sampling case (Morral et al., 2017; Koloskova et al., 2020; Li et al., 2022), which inherently contains the consensus error analyzed in Lemma 3.1. The details are deferred to Appedix E.1 – E.3. We require $\gamma_\star > 1/2\mu$ in the case of $a = 1$ to ensure that the largest eigenvalue of $\mathbf{M}$ is negative, as this is a necessary condition for the existence of $\mathbf{V}$ in (13) (otherwise the integration diverges). As a specific instance, when there is only one agent ($N = 1$), $\mathbf{V}$ and $\mathbf{V}'$ reduce to the matrices specified in the CLT result of vanilla SGD, e.g., Lemma 3.1 in Hu et al. (2022). By modifying (14) under different time scales and assuming $K_l = K$ for all $l$, we can recover the CLT result in Li et al. (2022) and the detailed discussion is included in Appendix F. Further elaboration on the implications of Theorem 3.3 is given next.

## 3.2 DISCUSSION

**Connection between CLT and Weighted MSE.** The asymptotic convergence rates for $\theta_n - \theta^*$ and $\bar{\theta}_n - \theta^*$ in Theorem 3.3 are $O(\sqrt{\gamma_n})$ and $O(1/\sqrt{n})$, respectively, and align with the rates in vanilla SGD (Borkar, 2009; Kushner & Yin, 2003; Fort, 2015), a special case $N = 1$ for GD-SGD. In addition, the interpretation of CLT in vanilla SGD for weighted MSE, as presented in Hu et al. (2022), can be extended to GD-SGD, e.g., for any weight vector $\mathbf{a} \in \mathbb{R}^d$ and large $n$, the weighted MSE can be approximated as $\mathbb{E}[\|\mathbf{a}^T(\theta_n - \theta^*)\|^2] = \mathbf{a}^T \mathbb{E}[(\theta_n - \theta^*)(\theta_n - \theta^*)^T]\mathbf{a} \approx \gamma_n \mathbf{a}^T \mathbf{V} \mathbf{a}$.

**Asymptotic Network Independence.** In D-SGD with constant doubly-stochastic $\mathbf{W}$, Olshevsky (2022) showed that after a transient period that depends on second largest eigenvalue modulus (SLEM) of $\mathbf{W}$, their finite-time bounds become independent of the underlying communication topology. This is also reflected in the CLT result of Corollary 1 in Morral et al. (2017), where the choice of $\mathbf{W}_n$ does not affect the limiting covariance matrix. Our Theorem 3.3 reaffirms this property in that the communication network does not influence the limiting covariance matrix in the asymptotic regime, extending Corollary 1 of Morral et al. (2017) for *i.i.d.* sampling into Markovian sampling and for more various types of $\{\mathbf{W}_n\}$. This implies that the effect of network topology is on the order of $o(\sqrt{\gamma_n})$ (or $o(1/\sqrt{n})$), which is consistent with the finite-time bound in Pu et al. (2020) such that the effect of network topology is on the higher order term. It diminishes faster than the dominant factor: *agent's sampling strategy*. One heuristic approach for enhancing the overall convergence can be to initially employ $\mathbf{W}$ with a smaller SLEM, as studied in Dimakis et al. (2010); Zhang (2020); Ye et al. (2022), to accelerate the mixing process. This is followed by a transition to sparse graphs, as investigated in Karakus et al. (2017); Neglia et al. (2020); Song et al. (2022), to reduce communication costs among agents.

**Linear Speedup.** Recent works (Koloskova et al., 2020; Khodadadian et al., 2022; Wang et al., 2023) have demonstrated that the factor $1/N$ is embedded in the dominant term of their convergence

bounds under both *i.i.d.* sampling and Markovian sampling, leading to linear speedup in the number of agents $N$. Our Theorem 3.3 also suggests this linear speed up in the number of agents. To see this, we first define matrices $\mathbf{V}_i$, $\mathbf{V}'_i$ of each agent $i$, i.e., with matrix $\mathbf{M}$ defined in Theorem 3.3,

$$\mathbf{V}_i = \int_0^\infty e^{\mathbf{M}t}\mathbf{U}_i e^{\mathbf{M}^T t}dt, \quad \mathbf{V}'_i = \mathbf{H}^{-1}\mathbf{U}_i\mathbf{H}^{-T}. \tag{15}$$

Then, we can decompose $\mathbf{V}$ and $\mathbf{V}'$ in Theorem 3.3 into

$$\mathbf{V} = \bar{\mathbf{V}}/N, \quad \mathbf{V}' = \bar{\mathbf{V}}'/N, \tag{16}$$

where $\bar{\mathbf{V}} = \sum_{i=1}^N \mathbf{V}_i/N, \bar{\mathbf{V}}' = \sum_{i=1}^N \mathbf{V}'_i/N$ represent the average limiting covariance matrices among $N$ agents. (16) implies the approximated weighted MSEs in the form of $\mathbf{a}^T\bar{\mathbf{V}}\mathbf{a}/N$ and $\mathbf{a}^T\bar{\mathbf{V}}'\mathbf{a}/N$, suggesting that the overall convergence will also be improved by $1/N$.

**Improvement on Sampling Strategy.** The SLEM-based technique has been widely used in the non-asymptotic analysis in the SGD, D-SGD and FL literature (Duchi et al., 2012; Sun et al., 2018; Zeng et al., 2022; Khodadadian et al., 2022), i.e., for each agent $i \in [N]$ and some constant $C > 0$,

$$\|F_i(\theta, X_n^i) - f_i(\theta)\| \leq C\|\theta\|\rho_i^n, \tag{17}$$

where $\rho_i$ represents the SLEM of the underlying Markov chain's transition matrix. However, recent works usually rely on the largest SLEM $\rho \triangleq \max_i \rho_i$, in other words, the worst-performing agent in their finite-time bounds (Wang et al., 2020a; Zeng et al., 2022; Khodadadian et al., 2022).

In contrast, as per (15), each agent holds its own limiting covariance matrices $\mathbf{V}_i$ and $\mathbf{V}'_i$, which are mainly determined by the matrix $\mathbf{U}_i$ that encapsulates the agent's sampling strategy $\{X_n^i\}$, and *contributes equally* to the overall performance of GD-SGD, as seen from (16). For any agent $i$, denote by $\mathbf{U}_i^X$ and $\mathbf{U}_i^Y$ the asymptotic covariance matrices associated with two candidate sampling strategies $\{X_n^i\}$ and $\{Y_n^i\}$, respectively. Let $\mathbf{V}^X$ and $\mathbf{V}^Y$ be the limiting covariance matrices in (13) when agent $i$ employs $\{X_n^i\}$ and $\{Y_n^i\}$, respectively, while keeping other agents' sampling strategies unchanged. Then, we have the following.

**Corollary 3.4.** *For agent $i$, if there exists two sampling strategies $\{X_n^i\}_{n\geq 0}$ and $\{Y_n^i\}_{n\geq 0}$ such that $\mathbf{U}_i^X \succeq \mathbf{U}_i^Y$, we have $\mathbf{V}^X \succeq \mathbf{V}^Y$.*

Corollary 3.4 directly follows from the definition of Loewner ordering, and Loewner ordering being closed under addition (i.e., $\mathbf{A} \succeq \mathbf{B}$ implies $\mathbf{A} + \mathbf{C} \succeq \mathbf{B} + \mathbf{C}$), as well as (16). It says, even if only one agent improves its sampling strategy from $\{X_n^i\}$ to $\{Y_n^i\}$, it leads to an overall reduction in $\mathbf{V}$ (in terms of Loewner ordering), thereby decreasing the weighted MSE and benefiting the entire group of $N$ agents. The subsequent question then arises: *How do we find an improved sampling strategy $\{Y_n^i\}$ over the baseline $\{X_n^i\}$ for each agent $i$?*

This question has been addressed by Hu et al. (2022), which investigates the 'efficiency ordering' of two sampling strategies in vanilla SGD (for a single agent). In particular, Theorem 3.6 (i) of Hu et al. (2022) shows that sampling strategy $\{Y_n\}$ is more efficient than $\{X_n\}$ if and only if $\boldsymbol{\Sigma}_X(\mathbf{g}) \succeq \boldsymbol{\Sigma}_Y(\mathbf{g})$ for any vector-valued function $\mathbf{g}(\cdot) \in \mathbb{R}^d$. Consequently, in the GD-SGD framework, if agent $i$ utilizes $\{Y_n^i\}$, which is more efficient than baseline $\{X_n^i\}$, it results in $\boldsymbol{\Sigma}_{X^i}(\nabla F_i(\theta^*, \cdot)) \succeq \boldsymbol{\Sigma}_{Y^i}(\nabla F_i(\theta^*, \cdot))$, that is, $\mathbf{U}_i^X \succeq \mathbf{U}_i^Y$ by definition. Corollary 3.4 then gives $\mathbf{V}^X \succeq \mathbf{V}^Y$ and implies the overall improvement in GD-SGD. For illustration purpose, we list a few examples stated in Hu et al. (2022) where two competing sampling strategies follow the efficiency ordering: i) When an agent has unrestricted access to the entire dataset, shuffling methods, including single shuffling and random reshuffling, are more efficient than *i.i.d.* sampling;[4] ii) When an agent is working with a graph-like data structure and employs a random walk, e.g., agent $i$ in Figure 1, using non-backtracking random walk (NBRW) is more efficient than simple random walk (SRW).[5] More examples of efficient MCMC samplers can be found in Haario et al. (2006); Yang & Rodríguez (2013).

---

[4]Random reshuffling refers to the repeated shuffling of a dataset after each complete traversal of the data points, while single shuffling only shuffles the dataset once and adheres to that specific order throughout the training process. In the L-SGD-FC framework, Yun et al. (2022) also proved that random reshuffling is better than *i.i.d* sampling via non-asymptotic analysis.

[5]Simple random walk refers to the walker that chooses one of the neighboring nodes uniformly at random. NBRW, as studied in Alon et al. (2007); Lee et al. (2012); Ben-Hamou et al. (2018) is a variation, which selects one of the neighbors uniformly at random, with the exception of the one visited in the last step.

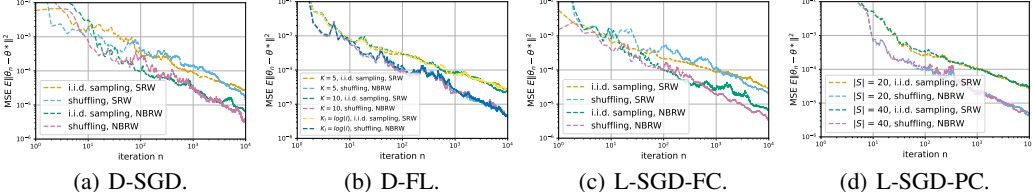

(a) D-SGD.          (b) D-FL.          (c) L-SGD-FC.          (d) L-SGD-PC.

Figure 2: Simulation results for logistic regression problem. (a) D-SGD with four combinations of sampling strategies. (b) D-FL with different communication intervals $K = 5, 10$ and $K_l = \log(l)$ ($l$: number of aggregations). (c) L-SGD-FC with $K = 10$ and four combinations of sampling strategies. (d) L-SGD-PC with without-replacement client sampling method and the client set size $|\mathcal{S}| = 20, 40$.

## 4 EXPERIMENTS

In this section, we empirically evaluate the consistency of our asymptotic results, presented in Section 3.2, i.e., the effect of sampling strategies of each agent, communication intervals and communication patterns in GD-SGD. We consider the $L_2$-regularized logistic regression problem

$$f(\theta) = \tfrac{1}{NB} \sum_{i=1}^{N} \sum_{j=1}^{B} \log(1 + \exp(-y_{i,j} \mathbf{x}_{i,j}^T \theta)) + \tfrac{1}{2} \|\theta\|^2, \tag{18}$$

where $\mathbf{x}_{i,j}, y_{i,j}$ are data point $j$ and its label held by agent $i$. We use the CIFAR-10 dataset (Krizhevsky & Hinton, 2009), which is distributed evenly to two groups of 50 agents ($N = 100$ agents in total) and each agent holds $B = 500$ data points. Each agent in the first group has full access to its entire dataset, thus can employ *i.i.d.* sampling or single shuffling. On the other hand, each agent in the other group has a graph-like structure and uses SRW or NBRW with reweighting to sample its local dataset with uniform weight, e.g., scenarios depicted in Figure 1.[6] We employ a decreasing step size $\gamma_n = n^{-0.9}$. Due to space constraints, we include more simulation result in Appendix G.

The simulation results illustrated in Figure 2 enter the asymptotic regime very early. All four subplots consistently demonstrate that a more efficient sampling strategy generally leads to a smaller MSE, e.g., single shuffling and NBRW outperform *i.i.d* sampling and SRW, even when only a subset of agents adopt improved strategies. This also suggests that efficient sampling strategies can achieve the same accuracy with fewer iterations. Note that purple and blue curves in Figures 2(a) and 2(c) take some time to outperform orange and green curves, respectively, which aligns with Ahn et al. (2020); Yun et al. (2022) because shuffling methods outperform *i.i.d.* sampling only after some period. Additionally, we observe in Figures 2(b) to 2(d) that communication intervals/patterns have minimal impact on MSE in the asymptotic regime (achieved by the straight line in the log-log scale). In Appendix G, we also present an experiment where the simulation does not enter the asymptotic regime early, causing communication intervals/patterns to affect MSE. However, employing more efficient sampling strategies can still help mitigate negative effects of less frequent aggregation.

## 5 CONCLUSION

In this work, we proposed a unified framework for the asymptotic analysis of GD-SGD. We demonstrated that individual agents can improve the overall performance of the distributed learning system by optimizing their sampling strategies. This is particularly relevant in large-scale machine learning, even when some agents are out of control or unavailable, which cannot be captured under finite-time upper bounds. In addition, we theoretically and empirically showed the limited impact on the convergence in the asymptotic regime under various communication patterns, as well as increasing communication interval, implying reduced communication costs. Future studies could pivot towards developing fine-grained finite-time bounds to individually characterize each agent's behavior, moving away from the current focus in the literature on establishing upper bounds predominantly associated with the worst-performing agent.

---

[6]Agents in both D-SGD and D-FL collaborate together to generate a deterministic communication matrix with Metropolis Hasting algorithm. The communication network among 100 agents and the graph-like structure of the dataset held by each agent are generated by *connected_watts_strogatz_graph()* in networkx package.

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

## A  DISCUSSION OF ASSUMPTION 2.3-II)

When $K_l \approx \log(l)$ (resp. $K_l \approx \log\log(l)$), as suggested by Li et al. (2022), it trivially satisfies $K_{\tau_n} = o(\gamma_n^{-1/2(L+1)}) = o(n^{1/2(L+1)})$ since by definition $K_{\tau_n} < K_n \approx \log(n)$ (resp. $\log\log(n)$), and $\log(n) = o(n^\epsilon)$ (resp. $\log\log(n) = o(n^\epsilon)$) for any $\epsilon > 0$. Besides, $\sum_n \eta_n^2 = \sum_n \gamma_n^2 K_{\tau_n}^{2(L+1)} \lesssim \sum_n n^{-2} n^{2(L+1)\epsilon} = \sum_n n^{2(L+1)\epsilon-2}$. To ensure $\sum_n \eta_n^2 < \infty$, it is sufficient to have $2(L+1)\epsilon - 2 < -1$, or equivalently, $\epsilon < 1/2(L+1)$. Since $\epsilon$ can be arbitrarily small to satisfy the condition, $\sum_n \eta_n^2 < \infty$ is satisfied.

When $K_l \approx \log(l)$, we can rewrite the last condition as

$$
\begin{aligned}
\frac{\eta_{n_l+1}}{\eta_{n_{l+1}+1}} &= \frac{\gamma_{n_l+1}}{\gamma_{n_{l+1}+1}} \frac{K_l^{L+1}}{K_{l+1}^{L+1}} \\
&= \left(\frac{n_{l+1}+1}{n_l+1}\right) \left(\frac{\log(l+1)+1}{\log(l)+1}\right)^{L+1} \\
&= \left(1 + \frac{K_{l+1}}{n_l+1}\right) \left(\frac{\log(l+1)+1}{\log(l)+1}\right)^{L+1},
\end{aligned}
\tag{19}
$$

where we have $n_l \approx \log(l!)$ such that $K_{l+1}/n_l \approx \log(l+1)/\log(l!) \to 0$ and $\log(l+1)/\log(l) \to 1$ as $l \to \infty$, which leads to $\lim_{n\to\infty} \eta_{n_l+1}/\eta_{n_{l+1}+1} = 1$. Similarly, for $K_l \approx \log\log(l)$, we have $n_l \approx \log(\prod_{s=1}^l \log(s))$ such that $K_{l+1}/n_l \approx \log\log(l+1)/\log\log(\prod_{s=1}^l \log(s)) \to 0$ and $\log\log(l+1)/\log\log(l) \to 1$ as $l \to \infty$, which also leads to $\lim_{n\to\infty} \eta_{n_l+1}/\eta_{n_{l+1}+1} = 1$.

## B  EXAMPLES OF COMMUNICATION MATRIX $\mathbf{W}$

### B.1  METROPOLIS HASTING ALGORITHM

In the decentralized learning such as D-SGD, HL-SGD and D-FL, $\mathbf{W}$ at the aggregation step can be generated locally using the Metropolis Hasting algorithm based on the underlying communication topology, and is deterministic (Pu et al., 2020; Koloskova et al., 2020; Zeng et al., 2022). Specifically, each agent $i$ exchanges its degree $d_i$ with its neighbors $j \in N(i)$, forming the weight

$$
\mathbf{W}(i,j) = \begin{cases} \min\{1/d_i, 1/d_j\}, & j \in N(i), \\ 1 - \sum_{j \neq N(i)} \mathbf{W}(i,j), & j = i. \end{cases}
\tag{20}
$$

In this case, $\mathbf{W}$ is doubly stochastic and symmetric. By Perron-Frobenius theorem, its SLEM $\lambda_2(\mathbf{W}) < 1$. Then, $\|\mathbf{W}^T\mathbf{W} - \mathbf{J}\| = \|\mathbf{W}^2 - \mathbf{J}\| = \lambda_2^2(\mathbf{W}) < 1$, which satisfies Assumption 2.5-ii).

### B.2  PARTIAL CLIENT SAMPLING IN FL

For L-SGD-FC studied in Stich (2018); Woodworth et al. (2020); Khodadadian et al. (2022), $\mathbf{W} = \mathbf{11}^T/N$ trivially satisfies Assumption 2.5-ii). For L-SGD-PC, on the other hand, only a small fraction of agents participate in each aggregation step for consensus (Li et al., 2020; Fraboni et al., 2022). Denote by $\mathcal{S}$ a randomly selected set of agents (without replacement) of fixed size $|\mathcal{S}| \in \{1, 2, \cdots, N\}$ at time $n$ and $\mathbf{W}_{\mathcal{S}}$ plays a role of aggregating $\theta_n^i$ for agent $i \in \mathcal{S}$, i.e., $\mathbf{W}_{\mathcal{S}}(i,j) = 1/|\mathcal{S}|$ for $i, j \in \mathcal{S}$, $\mathbf{W}_{\mathcal{S}}(i,i) = 1$ for $i \notin \mathcal{S}$, and $\mathbf{W}_{\mathcal{S}}(i,j) = 0$ otherwise. Additionally, the central server needs to broadcast updated parameter $\theta_{n+1}$ to the newly selected set $\mathcal{S}'$ with the same size, which results in a bijective mapping $\sigma$ (for $\mathcal{S} \to \mathcal{S}'$ and $[N]/\mathcal{S} \to [N]/\mathcal{S}'$) and a corresponding permutation matrix $\mathbf{T}_{\mathcal{S}\to\mathcal{S}'}$. Specifically, $\mathbf{T}_{\mathcal{S}\to\mathcal{S}'}(i,j) = 1$ if $j = \sigma(i)$ and $\mathbf{T}_{\mathcal{S}\to\mathcal{S}'}(i,j) = 0$ otherwise. Then, the communication matrix becomes $\mathbf{W} = \mathbf{T}_{\mathcal{S}\to\mathcal{S}'}\mathbf{W}_{\mathcal{S}}$. Note that $\mathbf{W}_{\mathcal{S}}$ is now a random matrix, since $\mathcal{S}$ is a randomly chosen subset of size $|\mathcal{S}|$. Clearly, for each choice of $\mathcal{S}$, $\mathbf{W}_{\mathcal{S}}$ is doubly stochastic, symmetric and $\mathbf{W}_{\mathcal{S}}^2 = \mathbf{W}_{\mathcal{S}}$. Taking the expectation of $\mathbf{W}_{\mathcal{S}}$ w.r.t the randomly selected set $\mathcal{S}$ gives

$$
\mathbb{E}_{\mathcal{S}}[\mathbf{W}_{\mathcal{S}}](i,j) = \begin{cases} (|\mathcal{S}| - 1)/N(N-1), & j \neq i, \\ 1 - (|\mathcal{S}| - 1)/N, & j = i, \end{cases}
\tag{21}
$$

for $i \in [N]$. Note that $\mathbb{E}_{\mathcal{S}}[\mathbf{W}_{\mathcal{S}}]$ has all positive entries. Therefore, we use the fact $\mathbf{T}^T \mathbf{T} = \mathbf{I}$ for a permutation matrix $\mathbf{T}$ such that $\|\mathbb{E}[\mathbf{W}^T \mathbf{W}] - \mathbf{J}\| = \|\mathbb{E}_{\mathcal{S}, \mathcal{S}'}[\mathbf{W}_{\mathcal{S}}^T \mathbf{T}_{\mathcal{S} \to \mathcal{S}'}^T \mathbf{T}_{\mathcal{S} \to \mathcal{S}'} \mathbf{W}_{\mathcal{S}}] - \mathbf{J}\| = \|\mathbb{E}_{\mathcal{S}}[\mathbf{W}_{\mathcal{S}}^T \mathbf{W}_{\mathcal{S}}] - \mathbf{J}\| = \|\mathbb{E}_{\mathcal{S}}[\mathbf{W}_{\mathcal{S}}] - \mathbf{J}\| < 1$ by Perron–Frobenius theorem and eigendecomposition, which satisfies Assumption 2.5-ii).

Next, we discuss the mathematical equivalence between our client sampling scheme and the commonly used FedAvg in the FL literature (Li et al., 2020; Fraboni et al., 2022) as follows:

1. At time $n$, the central server updates its global parameter $\theta_n = \frac{1}{|\mathcal{S}|} \sum_{i \in \mathcal{S}} \theta_n^i$ from the agents in the previous set $\mathcal{S}$. Then, the central server selects a new subset of agents $\mathcal{S}'$ and broadcasts $\theta_n$ to agent $i \in \mathcal{S}'$, i.e., $\theta_n^i = \theta_n$;

2. Each selected agent $i$ computes $K$ steps of SGD locally and consecutively updates its local parameter $\theta_{n+1}^i, \cdots, \theta_{n+K}^i$ according to (2a);

3. Each selected agent $i \in \mathcal{S}'$ uploads $\theta_{n+K}^i$ to the central server.

Then, the central server repeats the above three steps with $\theta_{n+K}$ and a new set of selected agents.

In our client sampling scheme, at the aggregation step $n$, the design of $\mathbf{W}_{\mathcal{S}}$ results in $\tilde{\theta}_n^i = \frac{1}{|\mathcal{S}|} \sum_{j \in \mathcal{S}} \theta_n^j$ for a selected agent $i \in \mathcal{S}$, and $\tilde{\theta}_n^i = \theta_n^i$ for an unselected agent $i \notin \mathcal{S}$. Meanwhile, the central server updates the global parameter $\tilde{\theta}_n = \tilde{\theta}_n^i$ for $i \in \mathcal{S}$. Then, the permutation matrix $\mathbf{T}_{\mathcal{S} \to \mathcal{S}'}$ ensures that the newly selected agent $i \in \mathcal{S}'$ will use $\tilde{\theta}_n$ as the initial point for its subsequent SGD iterations. Consequently, from the selected agents' perspective, the communication matrix $\mathbf{W} = \mathbf{T}_{\mathcal{S} \to \mathcal{S}'} \mathbf{W}_{\mathcal{S}}$ corresponds to step 1 in FedAvg. As we can observe, both algorithms update the global parameter identically from the central server's viewpoint, rendering them mathematically equivalent regarding the global parameter update.

We acknowledge that under the *i.i.d* sampling strategy, the behavior of unselected agents in our algorithm differs from FedAvg. Specifically, unselected agents are idle in FedAvg, while they continue the SGD computation in our algorithm (despite not contributing to the global parameter update). Importantly, when an unselected agent is later selected, the central server overwrites its local parameter during the broadcasting process. This ensures that the activities of agents when they are unselected have no impact on the global parameter update.

As far as we are aware, the FedAvg algorithm under the *Markovian* sampling strategy remains unexplored in the FL literature. Extrapolating the behavior of unselected agents in FedAvg from i.i.d sampling to Markovian sampling suggests that unselected agents would remain idle. In contrast, our algorithm enables unselected agents to continue evolving $X_n^i$. These additional transitions contribute to faster mixing of the Markov chain for each unselected agent and a smaller bias of $F_i(\theta, X_n^i)$ relative to its mean-field $f_i(\theta)$, potentially accelerating the convergence.

## C   PROOF OF LEMMA 3.1

Let $\mathbf{J}_\perp \triangleq \mathbf{I}_N - \mathbf{J} \in \mathbb{R}^{N \times N}$ and $\mathcal{J}_\perp \triangleq \mathbf{J}_\perp \otimes \mathbf{I}_d \in \mathbb{R}^{Nd \times Nd}$, where $\otimes$ is the Kronecker product. Let $\Theta_n = [\theta_n^1, \cdots, \theta_n^N]^T \in \mathbb{R}^{Nd}$. Then, motivated by Morral et al. (2017), we define a sequence $\phi_n \triangleq \eta_{n+1}^{-1} \mathcal{J}_\perp \Theta_n \in \mathbb{R}^{Nd}$ in the *increasing communication interval case* (resp. $\phi_n \triangleq \gamma_{n+1}^{-1} \mathcal{J}_\perp \Theta_n$ in the *bounded communication interval case*), where $\eta_{n+1}$ is defined in Assumption 2.3-ii). $\mathcal{J}_\perp \Theta_n = \Theta_n - \frac{1}{N}(\mathbf{1}\mathbf{1}^T \otimes \mathbf{I}_d) \Theta_n$ represents the consensus error of the model.

We first give the following lemma that shows the pathwise boundedness of $\phi_n$.

**Lemma C.1.** *Let Assumptions 2.1, 2.3, 2.4 and 2.5 hold. For any compact set $\Omega \subset \mathbb{R}^{Nd}$, the sequence $\phi_n$ satisfies*

$$\sup_n \mathbb{E}[\|\phi_n\|^2 \mathbb{1}_{\cap_{j \leq n-1}\{\Theta_j \in \Omega\}}] < \infty$$

*for both increasing and bounded communication interval cases.*

Lemma C.1 and Assumption 2.3 imply that for any $n \geq 0$, there exists some constant $C$ dependent on $\Omega$ such that

- increasing communication interval case:
$$\mathbb{E}[\|\mathcal{J}_\perp \Theta_n\|^2 \mathbb{1}_{\cap_{j \leq n-1}\{\Theta_j \in \Omega\}}] = \eta_{n+1}^2 \mathbb{E}[\|\phi_n\|^2 \mathbb{1}_{\cap_{j \leq n-1}\{\Theta_j \in \Omega\}}] \leq C\eta_{n+1}^2.$$

- bounded communication interval case:
$$\mathbb{E}[\|\mathcal{J}_\perp \Theta_n\|^2 \mathbb{1}_{\cap_{j \leq n-1}\{\Theta_j \in \Omega\}}] = \gamma_{n+1}^2 \mathbb{E}[\|\phi_n\|^2 \mathbb{1}_{\cap_{j \leq n-1}\{\Theta_j \in \Omega\}}] \leq C\gamma_{n+1}^2.$$

By Markov's inequality, we have

$$P(\|\mathcal{J}_\perp \Theta_n\| \mathbb{1}_{\cap_{j \leq n-1}\{\Theta_j \in \Omega\}} > \epsilon) \leq \frac{\mathbb{E}\left[\|\mathcal{J}_\perp \Theta_n\|^2 \mathbb{1}_{\cap_{j \leq n-1}\{\Theta_j \in \Omega\}}\right]}{\epsilon^2} \qquad (22)$$

for an arbitrary constant $\epsilon > 0$. Since $\{\gamma_n\}_{n \geq 0}$ and $\{\eta_n\}_{n \geq 0}$ are square-summable from Assumption 2.3, we have

$$\sum_{n=1}^\infty P(\|\mathcal{J}_\perp \Theta_n\| \mathbb{1}_{\cap_{j \leq n-1}\{\Theta_j \in \Omega\}} > \epsilon) < \infty.$$

Thus, from Borel-Cantelli lemma, we have

$$\lim_{n \to \infty} \mathcal{J}_\perp \Theta_n \mathbb{1}_{\cap_{j \leq n-1}\{\Theta_j \in \Omega\}} = \mathbf{0} \quad a.s.$$

Let $\{\Omega_m\}_{m \geq 0}$ be a sequence of increasing compact subset of $\mathbb{R}^{Nd}$ such that $\bigcup_m \Omega_m = \mathbb{R}^{Nd}$. Together with Lemma C.1, we know that for any $m \geq 0$,

$$\lim_{n \to \infty} \mathcal{J}_\perp \Theta_n \mathbb{1}_{\cap_{j \leq n-1}\{\Theta_j \in \Omega_m\}} = \mathbf{0} \quad a.s. \qquad (23)$$

(23) indicates either one of the following two cases:

- there exists some trajectory-dependent index $m'$ such that each trajectory $\{\Theta_n\}_{n \geq 0}$ is always within the compact set $\Omega_{m'}$, i.e., $\mathbb{1}_{\cap_{j \leq n}\{\Theta_j \in \Omega_{m'}\}} = 1$ (satisfied by the construction of increasing compact sets $\{\Omega_m\}_{m \geq 0}$ and Assumption 2.4), and we have $\lim_{n \to \infty} \mathcal{J}_\perp \Theta_n = \mathbf{0}$;
- $\Theta_n$ will escape the compact set $\Omega_m$ eventually for any $m \geq 0$ in finite time such that $\mathbb{1}_{\cap_{j \leq n-1}\{\Theta_j \in \Omega_m\}} = 0$ when $n$ is large enough.

We can see that the second case contradicts Assumption 2.4 because we assume every trajectory $\{\Theta_n\}_{n \geq 0}$ is within some compact set. Therefore, (23) for any $m \geq 0$ is equivalent to showing $\lim_{n \to \infty} \mathcal{J}_\perp \Theta_n = \mathbf{0}$, which completes the proof of Lemma 3.1.

*Proof of Lemma C.1.* We begin by rewriting the GD-SGD iterates in the matrix form,

$$\Theta_{n+1} = \mathcal{W}_n \left(\Theta_n - \gamma_{n+1} \nabla \mathbf{F}(\Theta_n, \mathbf{X}_n)\right), \qquad (24)$$

where $\mathbf{X}_n \triangleq (X_n^1, X_n^2, \cdots, X_n^N)$ and $\nabla \mathbf{F}(\Theta_n, \mathbf{X}_n) \triangleq [\nabla F_1(\theta_n^1, X_n^1)^T, \cdots, \nabla F_N(\theta_n^N, X_n^N)^T]^T \in \mathbb{R}^{Nd}$. Recall $\theta_n \triangleq \frac{1}{N}\sum_{i=1}^N \theta_n^i \in \mathbb{R}^d$ and we have $[\theta_n^T, \cdots, \theta_n^T]^T = \frac{1}{N}(\mathbf{1}\mathbf{1}^T \otimes \mathbf{I}_d)\Theta_n \in \mathbb{R}^{Nd}$.

**Case 1 (Increasing communication interval $K_{\tau_n}$):** By left multiplying (24) with $\frac{1}{N}(\mathbf{1}\mathbf{1}^T \otimes \mathbf{I}_d)$, along with $\gamma_{n+1} = \eta_{n+1}/K_{\tau_{n+1}}^{L+1}$ in Assumption 2.3-ii), we have the following iteration

$$\frac{1}{N}(\mathbf{1}\mathbf{1}^T \otimes \mathbf{I}_d)\Theta_{n+1} = \frac{1}{N}(\mathbf{1}\mathbf{1}^T \otimes \mathbf{I}_d)\Theta_n - \eta_{n+1}\frac{1}{N}(\mathbf{1}\mathbf{1}^T \otimes \mathbf{I}_d)\frac{\nabla \mathbf{F}(\Theta_n, \mathbf{X}_n)}{K_{\tau_{n+1}}^{L+1}}, \qquad (25)$$

where the equality comes from $\frac{1}{N}(\mathbf{1}\mathbf{1}^T \otimes \mathbf{I}_d)\mathcal{W}_n = \frac{1}{N}(\mathbf{1}\mathbf{1}^T\mathbf{W}_n \otimes \mathbf{I}_d) = \frac{1}{N}(\mathbf{1}\mathbf{1}^T \otimes \mathbf{I}_d)$. With (24) and (25), we have

$$\begin{aligned}
\Theta_{n+1} &- \frac{1}{N}(\mathbf{1}\mathbf{1}^T \otimes \mathbf{I}_d)\Theta_{n+1} \\
&= \left(\mathcal{W}_n - \frac{1}{N}(\mathbf{1}\mathbf{1}^T \otimes \mathbf{I}_d)\right)\Theta_n - \eta_{n+1}\left(\mathcal{W}_n - \frac{1}{N}(\mathbf{1}\mathbf{1}^T \otimes \mathbf{I}_d)\right)\frac{\nabla \mathbf{F}(\Theta_n, \mathbf{X}_n)}{K_{\tau_{n+1}}^{L+1}} \\
&= (\mathbf{J}_\perp \mathbf{W}_n \otimes \mathbf{I}_d)\mathcal{J}_\perp \Theta_n - \eta_{n+1}(\mathbf{J}_\perp \mathbf{W}_n \otimes \mathbf{I}_d)\frac{\nabla \mathbf{F}(\Theta_n, X_n)}{K_{\tau_{n+1}}^{L+1}} \\
&= \eta_{n+1}(\mathbf{J}_\perp \mathbf{W}_n \otimes \mathbf{I}_d)\left(\eta_{n+1}^{-1}\mathcal{J}_\perp \Theta_n - \frac{\nabla \mathbf{F}(\Theta_n, \mathbf{X}_n)}{K_{\tau_{n+1}}^{L+1}}\right),
\end{aligned} \qquad (26)$$

where the second equality comes from $\mathcal{W}_n - \frac{1}{N}(\mathbf{1}\mathbf{1}^T \otimes \mathbf{I}_d) = (\mathbf{W}_n - \frac{1}{N}\mathbf{1}\mathbf{1}^T) \otimes \mathbf{I}_d = \mathbf{J}_\perp \mathbf{W}_n \otimes \mathbf{I}_d$ and $(\mathbf{J}_\perp \mathbf{W}_n \otimes \mathbf{I}_d)\mathcal{J}_\perp = \mathbf{J}_\perp \mathbf{W}_n \mathbf{J}_\perp \otimes \mathbf{I}_d = \mathbf{J}_\perp \mathbf{W}_n \otimes \mathbf{I}_d$. Let $a_n \triangleq \eta_n/\eta_{n+1}$, dividing both sides of (26) by $\eta_{n+2}$ gives

$$\phi_{n+1} = a_{n+1}(\mathbf{J}_\perp \mathbf{W}_n \otimes \mathbf{I}_d)\left(\phi_n - \frac{\nabla \mathbf{F}(\Theta_n, \mathbf{X}_n)}{K_{\tau_{n+1}}^{L+1}}\right). \tag{27}$$

Define the filtration $\{\mathcal{F}_n\}_{n \geq 0}$ as $\mathcal{F}_n \triangleq \sigma\{\Theta_0, \mathbf{X}_0, \mathbf{W}_0, \Theta_1, \mathbf{X}_1, \mathbf{W}_1, \cdots, \mathbf{X}_{n-1}, \mathbf{W}_{n-1}, \Theta_n, \mathbf{X}_n\}$. Recursively computing (27) w.r.t the time interval $[n_l, n_{l+1}]$ gives

$$\phi_{n_{l+1}} = \left[\prod_{k=n_l+1}^{n_{l+1}} a_k\right]\left(\left[\mathbf{J}_\perp \prod_{k=n_l}^{n_{l+1}-1} \mathbf{W}_k\right]\otimes \mathbf{I}_d\right)\phi_{n_l} - \sum_{k=n_l}^{n_{l+1}-1}\left[\prod_{i=k+1}^{n_{l+1}} a_i\right]\left(\left[\mathbf{J}_\perp \prod_{i=k}^{n_{l+1}-1} \mathbf{W}_i\right]\otimes \mathbf{I}_d\right)\frac{\nabla \mathbf{F}(\Theta_k, \mathbf{X}_k)}{K_{l+1}^{L+1}}$$

$$= \frac{\eta_{n_l+1}}{\eta_{n_{l+1}+1}}(\mathbf{J}_\perp \mathbf{W}_{n_l} \otimes \mathbf{I}_d)\phi_{n_l} - \sum_{k=n_l}^{n_{l+1}-1}\frac{\eta_{n_l+1}}{\eta_{k+2}}(\mathbf{J}_\perp \mathbf{W}_{n_l} \otimes \mathbf{I}_d)\frac{\nabla \mathbf{F}(\Theta_k, \mathbf{X}_k)}{K_{l+1}^{L+1}}, \tag{28}$$

where $\prod$ is the backward multiplier, the second equality comes from $\mathbf{J}_\perp \mathbf{W}_n \mathbf{J}_\perp = \mathbf{J}_\perp \mathbf{W}_n$ and $\mathbf{W}_k = \mathbf{I}_N$ for $k \notin \{n_l\}$. In Assumption 2.5, we have $\|\mathbb{E}_{\mathbf{W} \sim \mathcal{P}_{n_l}}[\mathbf{W}^T \mathbf{J}_\perp \mathbf{W}]\| = \|\mathbb{E}_{\mathbf{W} \sim \mathcal{P}_{n_l}}[\mathbf{W}^T\mathbf{W} - \mathbf{J}]\| \leq C_1 < 1$. Then,

$$\mathbb{E}[\|\phi_{n_{l+1}}\|^2|\mathcal{F}_{n_l}]$$

$$= \left(\frac{\eta_{n_l+1}}{\eta_{n_{l+1}+1}}\right)^2 \phi_{n_l}^T \mathbb{E}_{\mathbf{W}_{n_l} \sim \mathcal{P}_{n_l}}\left[(\mathbf{J}_\perp \mathbf{W}_{n_l} \otimes \mathbf{I}_d)^T (\mathbf{J}_\perp \mathbf{W}_{n_l} \otimes \mathbf{I}_d)\right]\phi_{n_l}$$

$$- 2\mathbb{E}\left[\sum_{k=n_l}^{n_{l+1}-1}\frac{\eta_{n_l+1}^2}{\eta_{n_{l+1}+1}\eta_{k+2}}\phi_{n_l}^T(\mathbf{J}_\perp \mathbf{W}_{n_l} \otimes \mathbf{I}_d)^T(\mathbf{J}_\perp \mathbf{W}_{n_l} \otimes \mathbf{I}_d)\frac{\nabla \mathbf{F}(\Theta_k, \mathbf{X}_k)}{K_{l+1}^{L+1}}\bigg|\mathcal{F}_{n_l}\right]$$

$$+ \mathbb{E}\left[\left\|\sum_{k=n_l}^{n_{l+1}-1}\frac{\eta_{n_l+1}}{\eta_{k+2}}(\mathbf{J}_\perp \mathbf{W}_{n_l} \otimes \mathbf{I}_d)\frac{\nabla \mathbf{F}(\Theta_k, \mathbf{X}_k)}{K_{l+1}^{L+1}}\right\|^2\bigg|\mathcal{F}_{n_l}\right]$$

$$\leq \left(\frac{\eta_{n_l+1}}{\eta_{n_{l+1}+1}}\right)^2 \phi_{n_l}^T \mathbb{E}_{\mathbf{W}_{n_l} \sim \mathcal{P}_{n_l}}\left[(\mathbf{W}_{n_l}^T \mathbf{J}_\perp \mathbf{W}_{n_l} \otimes \mathbf{I}_d)\right]\phi_{n_l}$$

$$- 2\left(\frac{\eta_{n_l+1}}{\eta_{n_{l+1}+1}}\right)^2 \mathbb{E}\left[\sum_{k=n_l}^{n_{l+1}-1}\phi_{n_l}^T(\mathbf{W}_{n_l}^T \mathbf{J}_\perp \mathbf{W}_{n_l} \otimes \mathbf{I}_d)\frac{\nabla \mathbf{F}(\Theta_k, \mathbf{X}_k)}{K_{l+1}^{L+1}}\bigg|\mathcal{F}_{n_l}\right] \tag{29}$$

$$+ \left(\frac{\eta_{n_l+1}}{\eta_{n_{l+1}+1}}\right)^2 \mathbb{E}\left[\left\|(\mathbf{J}_\perp \mathbf{W}_{n_l} \otimes \mathbf{I}_d)\sum_{k=n_l}^{n_{l+1}-1}\frac{\nabla \mathbf{F}(\Theta_k, \mathbf{X}_k)}{K_{l+1}^{L+1}}\right\|^2\bigg|\mathcal{F}_{n_l}\right]$$

$$\leq \left(\frac{\eta_{n_l+1}}{\eta_{n_{l+1}+1}}\right)^2 C_1\|\phi_{n_l}\|^2 + 2\left(\frac{\eta_{n_l+1}}{\eta_{n_{l+1}+1}}\right)^2 C_1\|\phi_{n_l}\|\mathbb{E}\left[\left\|\sum_{k=n_l}^{n_{l+1}-1}\frac{\nabla \mathbf{F}(\Theta_k, \mathbf{X}_k)}{K_{l+1}^{L+1}}\right\|\bigg|\mathcal{F}_{n_l}\right]$$

$$+ \left(\frac{\eta_{n_l+1}}{\eta_{n_{l+1}+1}}\right)^2 C_1\mathbb{E}\left[\left\|\sum_{k=n_l}^{n_{l+1}-1}\frac{\nabla \mathbf{F}(\Theta_k, \mathbf{X}_k)}{K_{l+1}^{L+1}}\right\|^2\bigg|\mathcal{F}_{n_l}\right],$$

where the first inequality comes from $\mathbf{J}_\perp^T \mathbf{J}_\perp = \mathbf{J}_\perp$ and $\eta_{k+2} \geq \eta_{n_{l+1}+1}$ for $k \in [n_l, n_{l+1} - 1]$. Then, we analyze the norm of the gradient $\|\nabla \mathbf{F}(\Theta_k, \mathbf{X}_k)\|$ in the second term on the RHS of (29) conditioned on $\mathcal{F}_{n_l}$. By Assumption 2.4, we assume $\Theta_{n_l}$ is within some compact set $\Omega$ at time $n_l$ such that $\sup_{i \in [N], X^i \in \mathcal{X}_i} \nabla F_i(\theta_{n_l}^i, X^i) \leq C_\Omega$ for some constant $C_\Omega$. For $n = n_l + 1$ and any $\mathbf{X} \in \mathcal{X}_1 \times \mathcal{X}_2 \times \cdots \times \mathcal{X}_N$, we have

$$\|\nabla \mathbf{F}(\Theta_{n_l+1}, \mathbf{X})\| \leq \|\nabla \mathbf{F}(\Theta_{n_l+1}, \mathbf{X}) - \nabla \mathbf{F}(\Theta_{n_l}, \mathbf{X})\| + \|\nabla \mathbf{F}(\Theta_{n_l}, \mathbf{X})\|.$$

Considering $\|\nabla\mathbf{F}(\Theta_{n_l},\mathbf{X})\|$, we have

$$\sup_{\mathbf{X}}\|\nabla\mathbf{F}(\Theta_{n_l},\mathbf{X})\|^2 \leq \sum_{i=1}^{N}\sup_{X^i\in\mathcal{X}_i}\|\nabla F_i(\theta_{n_l}^i,X^i)\|^2 \leq NC_\Omega^2$$

such that $\|\nabla\mathbf{F}(\Theta_{n_l},\mathbf{X})\| \leq \sqrt{N}C_\Omega$. In addition, we have

$$
\begin{aligned}
\|\nabla\mathbf{F}(\Theta_{n_l+1},\mathbf{X}) - \nabla\mathbf{F}(\Theta_{n_l},\mathbf{X})\|^2 &= \sum_{i=1}^{N}\|\nabla F_i(\theta_{n_l+1}^i,X^i) - \nabla F_i(\theta_{n_l}^i,X^i)\|^2 \\
&\leq \sum_{i=1}^{N}L^2\|\theta_{n_l+1}^i - \theta_{n_l}^i\|^2 \\
&\leq \sum_{i=1}^{N}\gamma_{n_l+1}^2 L^2\|\nabla F_i(\theta_{n_l}^i,X_{n_l}^i)\|^2 \\
&\leq \gamma_{n_l+1}^2 L^2 NC_\Omega^2
\end{aligned}
\tag{30}
$$

such that $\|\nabla\mathbf{F}(\Theta_{n_l+1},\mathbf{X}) - \nabla\mathbf{F}(\Theta_{n_l},\mathbf{X})\| \leq \sqrt{N}\gamma_{n_l+1}LC_\Omega$. Thus, for any $\mathbf{X}$,

$$\|\nabla\mathbf{F}(\Theta_{n_l+1},\mathbf{X})\| \leq (1+\gamma_{n_l+1}L)\sqrt{N}C_\Omega. \tag{31}$$

For $n = n_l + 2$ and any $\mathbf{X}$, we have

$$\|\nabla\mathbf{F}(\Theta_{n_l+2},\mathbf{X})\| \leq \|\nabla\mathbf{F}(\Theta_{n_l+2},\mathbf{X}) - \nabla\mathbf{F}(\Theta_{n_l+1},\mathbf{X})\| + \|\nabla\mathbf{F}(\Theta_{n_l+1},\mathbf{X})\|.$$

Similar to the steps in (30), we have

$$
\begin{aligned}
\|\nabla\mathbf{F}(\Theta_{n_l+2},\mathbf{X}) - \nabla\mathbf{F}(\Theta_{n_l+1},\mathbf{X})\|^2 &\leq \sum_{i=1}^{N}\gamma_{n_l+2}^2 L^2\|\nabla F_i(\theta_{n_l+1}^i,X_{n_l+1}^i)\|^2 \\
&= \gamma_{n_l+2}^2 L^2\|\nabla\mathbf{F}(\Theta_{n_l+1},\mathbf{X}_{n_l+1})\|^2.
\end{aligned}
\tag{32}
$$

Then, $\|\nabla\mathbf{F}(\Theta_{n_l+2},\mathbf{X})\| \leq (1+\gamma_{n_l+2}L)\sup_{\mathbf{X}}\|\nabla\mathbf{F}(\Theta_{n_l+1},\mathbf{X})\|$ and, together with (31), we have

$$\|\nabla\mathbf{F}(\Theta_{n_l+2},\mathbf{X})\| \leq (1+\gamma_{n_l+2}L)(1+\gamma_{n_l+1}L)\sqrt{N}C_\Omega. \tag{33}$$

By induction, $\|\nabla\mathbf{F}(\Theta_{n_l+m},\mathbf{X})\| \leq \prod_{s=1}^{m}(1+\gamma_{n_l+s}L)\sqrt{N}C_\Omega$ for $m \in [1, K_{l+1}-1]$.

The next step is to analyze the growth rate of $\prod_{s=1}^{m}(1+\gamma_{n_l+s}L)$. By $1+x \leq e^x$ for $x \geq 0$, we have

$$\prod_{s=1}^{m}(1+\gamma_{n_l+s}L) \leq e^{L\sum_{s=1}^{m}\gamma_{n_l+s}}.$$

For step size $\gamma_n = 1/n$, we have $L\sum_{s=1}^{m}\gamma_{n_l+s} = L\sum_{s=1}^{m}1/(n_l+s) < L\sum_{s=1}^{m}1/s < L(\log(m)+1)$ such that $\prod_{s=1}^{m}(1+\gamma_{n_l+s}L) < (em)^L$. Then,

$$\left\|\sum_{k=n_l}^{n_{l+1}-1}\frac{\nabla\mathbf{F}(\Theta_k,\mathbf{X}_k)}{K_{l+1}^{L+1}}\right\| \leq \frac{1}{K_{l+1}^{L+1}}\sum_{k=n_l}^{n_{l+1}-1}\|\nabla\mathbf{F}(\Theta_k,\mathbf{X}_k)\| \leq \frac{1}{K_{l+1}^{L+1}}\sqrt{N}e^L C_\Omega\sum_{m=0}^{K_{l+1}-1}m^L \leq \sqrt{N}e^L C_\Omega, \tag{34}$$

where the last inequality comes from $\sum_{m=0}^{K_{l+1}-1}m^L < K_{l+1}(K_{l+1}-1)^L < K_{l+1}^{L+1}$. We can see the sum of the norm of the gradients are bounded by $\sqrt{N}e^L C_\Omega$, which only depends on the compact set $\Omega$ at time $n = n_l$.

Let $\delta_1 \in (C_1, 1)$. Since from Assumption 2.3-ii), $\lim_{l\to\infty}\eta_{n_l+1}/\eta_{n_{l+1}+1} = 1$, there exists some large enough $l_0$ such that $(\frac{\eta_{n_l+1}}{\eta_{n_{l+1}+1}})^2 C_1 < \delta_1 < \delta_2 := (\delta_1+1)/2 < 1$ for any $l > l_0$. Note that $\delta_1$ depends only on $C_1$ and is independent of $\mathcal{F}_n$. Then, let $\tilde{C}_\Omega := \sqrt{N}e^L C_\Omega$, we can rewrite (29) as

$$
\begin{aligned}
\mathbb{E}[\|\phi_{n_l+1}\|^2|\mathcal{F}_{n_l}] &\leq \delta_1\|\phi_{n_l}\|^2 + 2\delta_1\tilde{C}_\Omega\|\phi_{n_l}\| + \delta_1\tilde{C}_\Omega^2 \\
&\leq \delta_2\|\phi_{n_l}\|^2 + M_\Omega,
\end{aligned}
\tag{35}
$$

where $M_\Omega$ satisfies $M_\Omega > 8\tilde{C}_\Omega^2/(1-\delta_1) + \delta_1 \tilde{C}_\Omega^2$, which is derived from rearranging (35) as $M_\Omega \geq (\delta_1 - \delta_2)\|\phi_{n_l}\|^2 + 2\delta_1 \tilde{C}_\Omega \|\phi_{n_l}\| + \delta_1 \tilde{C}_\Omega^2$ and upper bounding the RHS. Upon noting that $\mathbb{1}_{\cap_{j \leq n_l}\{\Theta_j \in \Omega\}} \leq \mathbb{1}_{\cap_{j \leq n_{l-1}}\{\Theta_j \in \Omega\}}$, we obtain

$$\mathbb{E}\left[\|\phi_{n_{l+1}}\|^2 \mathbb{1}_{\cap_{j \leq n_l}\{\Theta_j \in \Omega\}}\right] \leq \delta_2 \mathbb{E}\left[\|\phi_{n_l}\|^2 \mathbb{1}_{\cap_{j \leq n_{l-1}}\{\Theta_j \in \Omega\}}\right] + M_\Omega. \tag{36}$$

The induction leads to $\mathbb{E}[\|\phi_{n_{l+1}}\|^2 \mathbb{1}_{\cap_{j \leq n_l}\{\Theta_j \in \Omega\}}] \leq \delta_2^{n_{l+1}-n_{l_0}} \mathbb{E}[\|\phi_{n_{l_0}}\|^2 \mathbb{1}_{\cap_{j \leq n_{l_0}-1}\{\Theta_j \in \Omega\}}] + M/(1-\delta_2) < \infty$ for any $l \geq l_0$. Besides, for $m \in (n_l, n_{l+1})$, by following the above steps (29) applied to (27), we have

$$\mathbb{E}[\|\phi_m\|^2|\mathcal{F}_{n_l}] \leq \left(\frac{\eta_{n_l+1}}{\eta_{m+1}}\right)^2 \|\phi_{n_l}\|^2 + 2\left(\frac{\eta_{n_l+1}}{\eta_{m+1}}\right)^2 \|\phi_{n_l}\| \mathbb{E}\left[\left\|\sum_{k=n_l}^{m-1} \frac{\nabla \mathbf{F}(\Theta_k, \mathbf{X}_k)}{K_{l+1}^{L+1}}\right\| \Big| \mathcal{F}_{n_l}\right]$$
$$+ \left(\frac{\eta_{n_l+1}}{\eta_{m+1}}\right)^2 \mathbb{E}\left[\left\|\sum_{k=n_l}^{m-1} \frac{\nabla \mathbf{F}(\Theta_k, \mathbf{X}_k)}{K_{l+1}^{L+1}}\right\|^2 \Big| \mathcal{F}_{n_l}\right]. \tag{37}$$

By (34) we already show that $\|\sum_{k=n_l}^{n_{l+1}-1} \frac{\nabla \mathbf{F}(\Theta_k, \mathbf{X}_k)}{K_{l+1}^{L+1}}\| < \infty$ conditioned on $\mathcal{F}_{n_l}$. Therefore, $\mathbb{E}[\|\phi_m\|^2 \mathbb{1}_{\cap_{j \leq n_l}\{\Theta_j \in \Omega\}}] < \infty$ for $m \in (n_l, n_{l+1})$. This completes the boundedness analysis of $\mathbb{E}[\|\phi_n\| \mathbb{1}_{\cap_{j \leq n-1}\{\Theta_j \in \Omega\}}]$.

**Case 2 (Bounded communication interval $K_{\tau_n} \leq K$):** In this case, we do not need the auxiliary step size $\eta_n$ and can directly work on $\gamma_n = 1/n^a$ for $a \in (0.5, 1]$. Similar to (26), we have

$$\Theta_{n+1} - \frac{1}{N}(\mathbf{1}\mathbf{1}^T \otimes \mathbf{I}_d)\Theta_{n+1} = \gamma_{n+1}(\mathbf{J}_\perp \mathbf{W}_n \otimes \mathbf{I}_d)\left(\gamma_{n+1}^{-1}\mathcal{J}_\perp \Theta_n - \nabla \mathbf{F}(\Theta_n, \mathbf{X}_n)\right), \tag{38}$$

and let $b_n \triangleq \gamma_n/\gamma_{n+1}$, dividing both sides of above equation by $\gamma_{n+2}$ gives

$$\phi_{n+1} = b_{n+1}(\mathbf{J}_\perp \mathbf{W}_n \otimes \mathbf{I}_d)(\phi_n - \nabla \mathbf{F}(\Theta_n, \mathbf{X}_n)). \tag{39}$$

Then, by following the similar steps in (28) and (29), we obtain

$$\mathbb{E}[\|\phi_{n_{l+1}}\|^2|\mathcal{F}_{n_l}] \leq \left(\frac{\gamma_{n_l+1}}{\gamma_{n_{l+1}+1}}\right)^2 C_1 \left(\|\phi_{n_l}\|^2 + 2\|\phi_{n_l}\| \mathbb{E}\left[\left\|\sum_{k=n_l}^{n_{l+1}-1} \nabla \mathbf{F}(\Theta_k, \mathbf{X}_k)\right\| \Big| \mathcal{F}_{n_l}\right]\right.$$
$$\left. + \mathbb{E}\left[\left\|\sum_{k=n_l}^{n_{l+1}-1} \nabla \mathbf{F}(\Theta_k, \mathbf{X}_k)\right\|^2 \Big| \mathcal{F}_{n_l}\right]\right). \tag{40}$$

Also similar to (31) - (34), we can bound the sum of the norm of the gradients as

$$\left\|\sum_{k=n_l}^{n_{l+1}-1} \nabla \mathbf{F}(\Theta_k, \mathbf{X}_k)\right\| \leq \sum_{k=n_l}^{n_{l+1}-1}\left[\prod_{s=n_l}^{k}(1+\gamma_{s+1}L)\right]\sqrt{N}C_\Omega. \tag{41}$$

Now that $K_l$ is bounded above by $K$, $\prod_{s=n_l}^{k}(1+\gamma_{s+1}L) \leq e^{L\sum_{s=n_l}^{k}\gamma_{s}+1} < e^{L\sum_{s=0}^{K-1}\gamma_{s+1}} := C_K$. Then, we further bound (41) as

$$\left\|\sum_{k=n_l}^{n_{l+1}-1} \nabla \mathbf{F}(\Theta_k, \mathbf{X}_k)\right\| \leq \sqrt{N}KC_KC_\Omega. \tag{42}$$

The subsequent proof is basically a replication of (35) - (37) and is therefore omitted. $\qquad \square$

## D    PROOF OF THEOREM 3.2

We focus on analyzing the convergence property of $\theta$, which is obtained by left multiplying (24) with $\frac{1}{N}(\mathbf{1}^T \otimes \mathbf{I}_d)$, i.e.,

$$\theta_{n+1} = \frac{1}{N}(\mathbf{1}^T \otimes \mathbf{I}_d)\Theta_{n+1} = \theta_n - \gamma_{n+1}\frac{1}{N}(\mathbf{1}^T \otimes \mathbf{I}_d)\nabla \mathbf{F}(\Theta_n, \mathbf{X}_n). \tag{43}$$

where the second equality comes from $\mathbf{W}_n$ being doubly stochastic and $\frac{1}{N}(\mathbf{1}^T \otimes \mathbf{I}_d)\mathcal{W}_n = \frac{1}{N}(\mathbf{1}^T\mathbf{W}_n \otimes \mathbf{I}_d) = \frac{1}{N}(\mathbf{1}^T \otimes \mathbf{I}_d)$.

For self-contained purpose, we first give the almost sure convergence result for the stochastic approximation that will be used in our proof.

**Theorem D.1** (Theorem 2 Delyon et al. (1999)). *Consider the stochastic approximation in the form of*

$$\theta_{n+1} = \theta_n + \gamma_{n+1}h(\theta_n) + \gamma_{n+1}e_{n+1} + \gamma_{n+1}r_{n+1}. \tag{44}$$

*Assume that*

    *C1. w.p.1, the closure of $\{\theta_n\}_{n\geq 0}$ is a compact subset of $\mathbb{R}^d$;*

    *C2. $\{\gamma_n\}$ is a decreasing sequence of positive number such that $\sum_n \gamma_n = \infty$;*

    *C3. w.p.1, $\lim_{p\to\infty} \sum_{n=1}^p \gamma_n e_n$ exists and is finite. Moreover, $\lim_{n\to\infty} r_n = 0$.*

    *C4. vector-valued function $h$ is continuous on $R^d$ and there exists a continuously differentiable function $V : \mathbb{R}^d \to \mathbb{R}$ such that $\langle \nabla V(\theta), h(\theta)\rangle \leq 0$ for all $\theta \in \mathbb{R}^d$. Besides, the interior of $V(\mathcal{L})$ is empty where $\mathcal{L} \triangleq \{\theta \in \mathbb{R}^d : \langle \nabla V(\theta), h(\theta)\rangle = 0\}$.*

*Then, w.p.1, $\limsup_n d(\theta_n, \mathcal{L}) = 0$.* $\qquad\qquad\square$

We can rewrite (43) as

$$
\begin{aligned}
\theta_{n+1} =& \theta_n - \gamma_{n+1}\frac{1}{N}(\mathbf{1}^T \otimes \mathbf{I}_d)\nabla\mathbf{F}(\Theta_n, \mathbf{X}_n)\\
=& \theta_n - \gamma_{n+1}\nabla f(\theta_n) - \gamma_{n+1}\left(\frac{1}{N}\sum_{i=1}^N \nabla f_i(\theta_n^i) - \nabla f(\theta_n)\right)\\
& - \gamma_{n+1}\left(\frac{1}{N}\sum_{i=1}^N \nabla F_i(\theta_n^i, X_n^i) - \frac{1}{N}\sum_{i=1}^N \nabla f_i(\theta_n^i)\right),
\end{aligned}
\tag{45}
$$

and work on the converging behavior of the third and fourth term. By definition of the mean-field function $\nabla f(\cdot)$, we have

$$r_n^{(A)} \triangleq \frac{1}{N}\sum_{i=1}^N \nabla f_i(\theta_n^i) - \nabla f(\theta_n) = \frac{1}{N}\sum_{i=1}^N \left[\nabla f_i(\theta_n^i) - \nabla f_i(\theta_n)\right]. \tag{46}$$

Additionally, we let

$$r_n^{(B)} \triangleq \frac{1}{N}\sum_{i=1}^N \nabla F_i(\theta_n^i, X_n^i) - \frac{1}{N}\sum_{i=1}^N \nabla f_i(\theta_n^i).$$

### D.1   ANALYSIS ON SEQUENCE $\{r_n^{(A)}\}$

By Lipschitz continuity of function $\nabla F_i(\cdot, X)$, we have

$$\left\|r_n^{(A)}\right\| \leq \frac{1}{N}\sum_{i=1}^N L\|\theta_n^i - \theta_n\| \leq \frac{L}{\sqrt{N}}\left\|\Theta_n - \frac{1}{N}(\mathbf{1}\mathbf{1}^T \otimes \mathbf{I}_d)\Theta_n\right\| = \frac{L}{\sqrt{N}}\|\mathcal{J}_\perp\Theta_n\|, \tag{47}$$

where the second inequality comes from the Cauchy-Schwartz inequality. In Appendix C, we have shown $\lim_n \mathcal{J}_\perp\Theta_n = \mathbf{0}$ almost surely such that $\lim_{n\to\infty} r_n^{(A)} = 0$ almost surely.

### D.2   ANALYSIS ON SEQUENCE $\{r_n^{(B)}\}$

Next, we further decompose $r_n^{(B)}$ in (45). For an ergodic transition matrix $\mathbf{P}$ and a function $v$ associated with the same state space $\mathcal{X}$, define the operator

$$\mathbf{P}^k v(x) \triangleq \sum_{y\in\mathcal{X}} \mathbf{P}^k(x, y)v(y)$$

for the $k$-step transition probability $\mathbf{P}^k(x, y)$. Denote by $\mathbf{P}_1, \cdots, \mathbf{P}_N$ the underlying transition matrices of all $N$ agents with corresponding stationary distribution $\boldsymbol{\pi}_1, \cdots, \boldsymbol{\pi}_N$. Then, for every function $\nabla F_i(\theta^i, \cdot) : \mathcal{X}_i \to \mathbb{R}^d$, there exists a corresponding Lipschitz-continuous function $m_{\theta^i}(\cdot) : \mathcal{X}_i \to \mathbb{R}^d$ such that

$$m_{\theta^i}(x) - \mathbf{P}_i m_{\theta^i}(x) = \nabla F_i(\theta^i, x) - \nabla f_i(\theta^i). \tag{48}$$

We defer the discussion about function $m_{\theta^i}(\cdot)$ and its Lipschitz property later in Section D.3.

Now with (48) we can decompose $\nabla F_i(\theta_n^i, X_n^i) - \nabla f_i(\theta_n^i)$ as

$$
\begin{aligned}
\nabla F_i(\theta_n^i, X_n^i) - \nabla f_i(\theta_n^i) = & m_{\theta_n^i}(X_n^i) - \mathbf{P}_i m_{\theta_n^i}(X_n^i) \\
= & \underbrace{m_{\theta_n^i}(X_n^i) - \mathbf{P}_i m_{\theta_n^i}(X_{n-1}^i)}_{e_{n+1}^i} \\
& + \underbrace{\mathbf{P}_i m_{\theta_n^i}(X_{n-1}^i)}_{\nu_n^i} - \underbrace{\mathbf{P}_i m_{\theta_{n+1}^i}(X_n^i)}_{\nu_{n+1}^i} \\
& + \underbrace{\mathbf{P}_i m_{\theta_{n+1}^i}(X_n^i) - \mathbf{P}_i m_{\theta_n^i}(X_n^i)}_{\xi_{n+1}^i}.
\end{aligned}
\tag{49}
$$

Here $\{\gamma_n e_n^i\}$ is a Martingale difference sequence and we need the martingale convergence theorem in Theorem D.2 as follows.

**Theorem D.2** (Theorem 6.4.6 Ross et al. (1996)). *For an $\mathcal{F}_n$-Martingale $S_n$, set $X_{n-1} = S_n - S_{n-1}$. If for some $1 \leq p \leq 2$,*

$$\sum_{n=1}^{\infty} \mathbb{E}[\|X_{n-1}\|^p | \mathcal{F}_{n-1}] < \infty \quad a.s. \tag{50}$$

*then $S_n$ converges almost surely.* □

We want to show that $\sum_n \gamma_{n+1}^2 \mathbb{E}[\|e_{n+1}^i\|^2 | \mathcal{F}_n] < \infty$ such that $\sum_n \gamma_n e_n^i$ converges almost surely by Theorem D.2. As we will later see in (60), with Lemma D.3 and Assumption 2.4, for a sample path ($\Theta_n$ within a compact set $\Omega$), $\sup_n \|m_{\theta_n^i}(x)\| < \infty$ and $\sup_n \|\mathbf{P}_i m_{\theta_n^i}(x)\| < \infty$ almost surely for all $x \in \mathcal{X}_i$. This ensures that $e_{n+1}^i$ is an $L_2$-bounded martingale difference sequence, i.e., $\sup_n \|e_{n+1}^i\| \leq \sup_n(\|m_{\theta_n^i}(X_{n+1}^i)\| + \|\mathbf{P}_i m_{\theta_n^i}(X_n^i)\|) \leq D_\Omega < \infty$. Together with Assumption 2.3, we get

$$\sum_n \gamma_{n+1}^2 \mathbb{E}[\|e_{n+1}^i\|^2 | \mathcal{F}_n] \leq D_\Omega \sum_n \gamma_{n+1}^2 < \infty \quad a.s. \tag{51}$$

and thus $\sum_n \gamma_n e_n^i$ converges almost surely.

Next, for the term $\nu_n^i$, by Abel transformation, we have

$$\sum_{k=0}^p \gamma_{k+1}(\nu_k^i - \nu_{k+1}^i) = \sum_{k=0}^p (\gamma_{k+1} - \gamma_k)\nu_k^i + \gamma_0 \nu_0^i - \gamma_{p+1} \nu_{p+1}^i. \tag{52}$$

As previously mentioned, for a given sample path, there always exists a compact subset such that $\|\mathbf{P}_i m_{\theta_n^i}(x)\|$ is bounded for all $n$ and $x \in \mathcal{X}^i$ such that $\sup_n \|\nu_n^i\| < \infty$ almost surely. Since $\lim_{n \to \infty}(\gamma_{n+1} - \gamma_n) = 0$, we have $\lim_{n \to \infty}(\gamma_{n+1} - \gamma_n)\nu_n^i = 0$. Note that there exists a path-dependent constant $C$ (that bounds $\|\nu_n^i\|$) such that for any $n \geq m$,

$$\left\| \sum_{k=m}^n (\gamma_{k+1} - \gamma_k)\nu_k^i \right\| \leq C \sum_{k=m}^n (\gamma_k - \gamma_{k+1}) = C(\gamma_m - \gamma_{n+1}) < C\gamma_m. \tag{53}$$

Since $\lim_{n \to \infty} \gamma_n = 0$, there exists a positive integer $M$ such that for all $n \geq m \geq M$, $\gamma_m < \epsilon/C$ and $\|\sum_{k=m}^n (\gamma_{k+1} - \gamma_k)\nu_k^i\| < \epsilon$ for every $\epsilon > 0$. Therefore, by definition $\{\sum_{k=0}^p (\gamma_{k+1} - \gamma_k)\nu_k^i\}_{p \geq 0}$ is a Cauchy sequence and $\sum_{k=0}^\infty (\gamma_{k+1} - \gamma_k)\nu_k^i$ converges by Cauchy convergence criterion. The last term of (52) tends to zero. Therefore, $\sum_{k=0}^\infty \gamma_{k+1}(\nu_k^i - \nu_{k+1}^i)$ converges and is finite.

For the last term $\xi_n^i$, Lemma D.3 leads to

$$\frac{1}{N} \sum_{i=1}^N \left\| \xi_{n+1}^i \right\| \le \frac{C'}{N} \sum_{i=1}^N \left\| \theta_{n+1}^i - \theta_n^i \right\| \le \frac{C'}{\sqrt{N}} \| \Theta_{n+1} - \Theta_n \|. \tag{54}$$

for the Lipschitz constant $C'$ of $\mathbf{P}_i m_{\theta^i}(x)$. However, the relationship between $\theta_n$ and $\theta_{n+1}$ is not obvious in the D-SGD and FL setting due to the update rule (24) with communication matrix $\mathcal{W}_n$, unlike the classical stochastic approximation shown in (44). We come up with the novel decomposition of $\xi_n^i$, which takes the consensus error into account, to solve this issue, i.e.,

$$\begin{aligned}
\xi_{n+1}^i = & \left[ \mathbf{P}_i m_{\theta_{n+1}^i}(X_n^i) - \mathbf{P}_i m_{\theta_{n+1}}(X_n^i) \right] + \left[ \mathbf{P}_i m_{\theta_{n+1}}(X_n^i) - \mathbf{P}_i m_{\theta_n}(X_n^i) \right] \\
& + \left[ \mathbf{P}_i m_{\theta_n}(X_n^i) \mathbf{P}_i m_{\theta_n^i}(X_n^i) \right].
\end{aligned} \tag{55}$$

Now, we have

$$\begin{aligned}
\frac{1}{N} \sum_{i=1}^N \left\| \xi_{n+1}^i \right\| \le & \frac{C'}{N} \sum_{i=1}^N \left( \left\| \theta_{n+1}^i - \theta_{n+1} \right\| + \left\| \theta_{n+1} - \theta_n \right\| + \left\| \theta_n - \theta_n^i \right\| \right) \\
\le & \frac{C'}{\sqrt{N}} \left\| \Theta_{n+1} - \frac{1}{N} \left( \mathbf{1} \mathbf{1}^T \otimes \mathbf{I}_d \right) \Theta_{n+1} \right\| + C' \| \theta_{n+1} - \theta_n \| \\
& + \frac{C'}{\sqrt{N}} \left\| \Theta_n - \frac{1}{N} \left( \mathbf{1} \mathbf{1}^T \otimes \mathbf{I}_d \right) \Theta_n \right\| \\
= & \frac{C'}{\sqrt{N}} \left( \| \mathcal{J}_\perp \Theta_{n+1} \| + \| \mathcal{J}_\perp \Theta_n \| \right) + C' \| \theta_{n+1} - \theta_n \| \\
= & \frac{C'}{\sqrt{N}} \left( \| \mathcal{J}_\perp \Theta_{n+1} \| + \| \mathcal{J}_\perp \Theta_n \| \right) + C' \gamma_{n+1} \left\| \frac{1}{N} (\mathbf{1}^T \otimes \mathbf{I}_d) \nabla \mathbf{F}(\Theta_n, \mathbf{X}_n) \right\|.
\end{aligned} \tag{56}$$

In Appendix C we have shown $\lim_{n \to \infty} \mathcal{J}_\perp \Theta_n = \mathbf{0}$ almost surely. Moreover, $\| \frac{1}{N} (\mathbf{1}^T \otimes \mathbf{I}_d) \nabla \mathbf{F}(\Theta_n, \mathbf{X}_n) \|$ is bounded per sample path. Therefore, $\lim_{n \to \infty} \frac{1}{N} \sum_{i=1}^N \| \xi_{n+1}^i \| = 0$ such that $\lim_{n \to \infty} \frac{1}{N} \sum_{i=1}^N \xi_{n+1}^i = 0$ almost surely.

To sum up, we decompose (45) into

$$\theta_{n+1} = \theta_n - \gamma_{n+1} \nabla f(\theta_n) - \gamma_{n+1} r_n^{(A)} - \gamma_{n+1} \frac{1}{N} \sum_{i=1}^N \left( e_{n+1}^i + \nu_n^i - \nu_{n+1}^i + \xi_{n+1}^i \right). \tag{57}$$

Now that $\lim_{p \to \infty} \sum_{n=1}^p \frac{1}{N} \sum_{i=1}^N \gamma_n e_n^i$ and $\lim_{p \to \infty} \sum_{n=0}^p \frac{1}{N} \sum_{i=1}^N \gamma_{n+1} (\nu_n^i - \nu_{n+1}^i)$ converge and are finite, $\lim_{n \to \infty} r_n^{(A)} = 0$, $\lim_{n \to \infty} \frac{1}{N} \sum_{i=1}^N \xi_n^i = 0$, all the conditions of C3 in Theorem D.1 are satisfied. Additionally, Assumption 2.4 corresponds to C1, Assumption 2.3 meets C2, and C4 is automatically satisfied when we choose the Lyapunov function $V(\theta) = f(\theta)$ and Assumption 2.1. Therefore, $\limsup_n \inf_{\theta^* \in \mathcal{L}} \| \theta_n - \theta^* \| = 0$.

### D.3 Discussion about Function $m_{\theta^i}(\cdot)$

The solution of the Poisson equation (48) has been studied in the literature, e.g., Kushner & Yin (2003); Chen et al. (2020); Hu et al. (2022). For self-contained purpose, we include the derivation of the closed-form $m_{\theta^i}(x)$ from scratch. Let $\nabla \mathbf{F}_i(\theta^i) \triangleq [\nabla F_i(\theta^i, 1), \cdots, \nabla F_i(\theta^i, |\mathcal{X}_i|)] \in \mathbb{R}^{d \times |\mathcal{X}_i|}$ and recall that $\mathbf{P}_i \in \mathbb{R}^{|\mathcal{X}_i| \times |\mathcal{X}_i|}$. We use $[\mathbf{A}]_{:,i}$ to denote the $i$-th column of matrix $\mathbf{A}$. For each agent $i \in \mathcal{N}$, we can obtain function $m_{\theta^i}(x)$ in the infinite sum form as follows.

$$\begin{aligned}
m_{\theta^i}(x) = & \sum_{k=0}^\infty \left( \left[ \nabla \mathbf{F}_i(\theta^i) (\mathbf{P}_i^k)^T \right]_{[:,x]} - \nabla f_i(\theta^i) \right) \\
= & \sum_{k=0}^\infty \left[ \nabla \mathbf{F}_i(\theta^i) \left( (\mathbf{P}_i^k)^T - \boldsymbol{\pi}_i \mathbf{1}^T \right) \right]_{[:,x]}.
\end{aligned} \tag{58}$$

Additionally,

$$\mathbf{P}_i m_{\theta^i}(x) = \sum_{k=1}^{\infty} \left[ \nabla \mathbf{F}_i(\theta^i) \left( (\mathbf{P}_i^k)^T - \boldsymbol{\pi}_i \mathbf{1}^T \right) \right]_{[:,x]}.$$

Therefore, the form of $m_{\theta^i}(x)$ in (58) satisfies the Poisson equation (48). Note that by induction we can get

$$\mathbf{P}_i^k - \mathbf{1}(\boldsymbol{\pi}_i)^T = \left( \mathbf{P}_i - \mathbf{1}(\boldsymbol{\pi}_i)^T \right)^k, \forall k \in \mathbb{N}, k \geq 1. \tag{59}$$

Then, we can further simplify (58) so that the closed-form expression of $m_{\theta^i}(x)$ is given by

$$
\begin{aligned}
m_{\theta^i}(x) &= \sum_{k=0}^{\infty} \left[ \nabla \mathbf{F}_i(\theta^i) \left( (\mathbf{P}_i)^T - \boldsymbol{\pi}_i \mathbf{1}^T \right)^k \right]_{[:,x]} - \nabla f_i(\theta^i) \\
&= \left[ \nabla \mathbf{F}_i(\theta^i) \sum_{k=0}^{\infty} \left( (\mathbf{P}_i)^T - \boldsymbol{\pi}_i \mathbf{1}^T \right)^k \right]_{[:,x]} - \nabla f_i(\theta^i) \\
&= \left[ \nabla \mathbf{F}_i(\theta^i) \left( \mathbf{I} - \mathbf{P}_i + \mathbf{1}(\boldsymbol{\pi}_i)^T \right)^{-1} \right]_{[:,x]} - \nabla f_i(\theta^i) \\
&= \sum_{y \in \mathcal{X}_i} \left( \mathbf{I} - \mathbf{P}_i + \mathbf{1}(\boldsymbol{\pi}_i)^T \right)^{-1}(x,y) \nabla F_i(\theta^i,y) - \nabla f_i(\theta^i),
\end{aligned}
\tag{60}
$$

where the first equality comes from (59). Since function $\nabla F_i$ is Lipschitz continuous, we have the following lemma.

**Lemma D.3.** *Under assumption (A1), functions $m_{\theta^i}(x)$ and $\mathbf{P}_i m_{\theta^i}(x)$ are both Lipschitz continuous in $\theta^i$ for any $x \in \mathcal{X}_i$.*

*Proof.* By (60), for any $\theta_1^i, \theta_2^i \in \mathbb{R}^d$ and $x \in \mathcal{X}_i$, we have

$$
\begin{aligned}
\left\| m_{\theta_1^i}(x) - m_{\theta_2^i}(x) \right\| &\leq \left\| \sum_{y \in \mathcal{X}_i} \left( \mathbf{I} - \mathbf{P}_i + \mathbf{1}(\boldsymbol{\pi}_i)^T \right)^{-1}(x,y) \left[ \nabla F_i(\theta_1^i,y) - \nabla F_i(\theta_2^i,y) \right] \right\| \\
&\quad + \left\| \nabla f_i(\theta_1^i) - \nabla f_i(\theta_2^i) \right\| \\
&\leq C_i \max_{y \in \mathcal{X}_i} \left\| \nabla F_i(\theta_1^i,y) - \nabla F_i(\theta_2^i,y) \right\| + \left\| \nabla f_i(\theta_1^i) - \nabla f_i(\theta_2^i) \right\| \\
&\leq (C_i L + 1)\|\theta_1^i - \theta_2^i\|,
\end{aligned}
\tag{61}
$$

where the second inequality holds for a constant $C_i$ that is the largest absolute value of the entry in the matrix $(\mathbf{I} - \mathbf{P}_i + \mathbf{1}(\boldsymbol{\pi}_i)^T)^{-1}$. Therefore, $m_{\theta^i}(x)$ is Lipschitz continuous in $\theta^i$. Moreover, following the similar steps as above, we have

$$
\begin{aligned}
\left\| \mathbf{P}_i m_{\theta_1^i}(x) - \mathbf{P}_i m_{\theta_2^i}(x) \right\| &= \left\| \sum_{y \in \mathcal{X}_i} \mathbf{P}_i(x,y) m_{\theta_1^i}(y) - \sum_{y \in \mathcal{X}_i} \mathbf{P}_i(x,y) m_{\theta_2^i}(y) \right\| \\
&= \left\| \sum_{y \in \mathcal{X}_i} \mathbf{P}_i(x,y) \left( m_{\theta_1^i}(y) - m_{\theta_2^i}(y) \right) \right\| \\
&\leq \sum_{y \in \mathcal{X}^i} \mathbf{P}_i(x,y) \left\| m_{\theta_1^i}(y) - m_{\theta_2^i}(y) \right\| \\
&\leq \max_{y \in \mathcal{X}^i} \left\| m_{\theta_1^i}(y) - m_{\theta_2^i}(y) \right\| \\
&\leq (C_i L + 1)\|\theta_1^i - \theta_2^i\|
\end{aligned}
\tag{62}
$$

such that $\mathbf{P}_i m_{\theta^i}(x)$ is also Lipschitz continuous in $\theta^i$, which completes the proof. $\square$

# E  PROOF OF THEOREM 3.3

To obtain Theorem 3.3, we need to utilize the existing CLT result for general SA in Theorem E.1 and check all the necessary conditions therein.

**Theorem E.1** (Theorem 2.1 Fort (2015)). *Consider the stochastic approximation iteration* (44)*, assume*

C1. *Let $\theta^*$ be the root of function $h$, i.e., $h(\theta^*) = 0$, and assume $\lim_{n\to\infty} \theta_n = \theta^*$. Moreover, assume the mean field $h$ is twice continuously differentiable in a neighborhood of $\theta^*$, and the Jacobian $\mathbf{H} \triangleq \nabla h(\theta^*)$ is Hurwitz, i.e., the largest real part of its eigenvalues $B < 0$;*

C2. *The step size $\sum_n \gamma_n = \infty$, $\sum_n \gamma_n^2 < \infty$, and either (i). $\log(\gamma_{n-1}/\gamma_n) = o(\gamma_n)$, or (ii). $\log(\gamma_{n-1}/\gamma_n) \sim \gamma_n/\gamma_\star$ for some $\gamma_\star > 1/2|B|$;*

C3. *$\sup_n \|\theta_n^i\| < \infty$ almost surely for any $i \in [N]$;*

C4. *(a) $\{e_n\}_{n\geq 0}$ is an $\mathcal{F}_n$-Martingale difference sequence, i.e., $\mathbb{E}[e_n|\mathcal{F}_{n-1}] = 0$, and there exists $\tau > 0$ such that $\sup_{n\geq 0} \mathbb{E}[\|e_n\|^{2+\tau}|\mathcal{F}_{n-1}] < \infty$;*

*(b) $\mathbb{E}[e_{n+1}e_{n+1}^T|\mathcal{F}_n] = \mathbf{U} + \mathbf{D}_n^{(A)} + \mathbf{D}_n^{(B)}$, where $\mathbf{U}$ is a symmetric positive semi-definite matrix and*

$$
\begin{cases}
\mathbf{D}_n^{(A)} \to 0 \quad \text{almost surely,} \\
\lim_n \gamma_n \mathbb{E}\left[\left\|\sum_{k=1}^n \mathbf{D}_k^{(B)}\right\|\right] = 0.
\end{cases}
\tag{63}
$$

C5. *Let $r_n = r_n^{(1)} + r_n^{(2)}$, $r_n$ is $\mathcal{F}_n$-adapted, and*

$$
\begin{cases}
\left\|r_n^{(1)}\right\| = o(\sqrt{\gamma_n}) \quad a.s. \\
\sqrt{\gamma_n}\left\|\sum_{k=1}^n r_k^{(2)}\right\| = o(1) \quad a.s.
\end{cases}
\tag{64}
$$

*Then,*

$$
\frac{1}{\sqrt{\gamma_n}}(\theta_n - \theta^*) \xrightarrow[n\to\infty]{dist.} \mathcal{N}(0, \mathbf{V}),
\tag{65}
$$

*where*

$$
\begin{cases}
\mathbf{V}\mathbf{H}^T + \mathbf{H}\mathbf{V} = -\mathbf{U} & \text{in case C2 (i),} \\
\mathbf{V}(\mathbf{I}_d + 2\gamma_\star\mathbf{H}^T) + (\mathbf{I}_d + 2\gamma_\star\mathbf{H})\mathbf{V} = -2\gamma_\star\mathbf{U} & \text{in case C2 (ii).}
\end{cases}
\tag{66}
$$

$\square$

Note that the matrix $\mathbf{U}$ in the condition C4(b) of Theorem E.1 was assumed to be positive definite in the original Theorem 2.1 Fort (2015). It was only to ensure that the solution $\mathbf{V}$ to the Lyapunov equation (66) is positive definite, which was only used for the stability of the related autonomous linear ODE (e.g., Theorem 3.16 Chellaboina & Haddad (2008) or Theorem 2.2.3 Horn & Johnson (1991)). However, in this paper, we do not need strict positive definite matrix $\mathbf{V}$. Therefore, we extend $\mathbf{U}$ to be positive semi-definite such that $\mathbf{V}$ is also positive semi-definite (see Lemma E.2 for the closed form of matrix $\mathbf{V}$). Such kind of extension does not change any of the proof steps in Fort (2015).

## E.1  DISCUSSION ABOUT C1-C3

Our Assumption 2.1 corresponds to C1 by letting function $h(\theta) = -\nabla f(\theta)$ therein. We can also let $\gamma_\star$ in Theorem 3.3 large enough to satisfy C2. The typical form of step size, also indicated in Fort (2015), is polynomial step size $\gamma_n \sim \gamma_\star/n^a$ for $a \in (0.5, 1]$. Note that $a \in (0.5, 1)$ satisfies C2 (i) and $a = 1$ satisfies C2 (ii). Assumption 2.4 corresponds to C3.[7]

---

[7] Theorem E.1 is slightly modified in terms of condition C3, which is mentioned as a special case in Section 2.2 Fort (2015). For the sake of mathematical simplicity, we stick to condition C3 in the proof.

### E.2  ANALYSIS OF C4

To check condition C4, we need to analyze the Martingale difference sequence $\{e_n^i\}$. Recall $e_{n+1}^i = m_{\theta_n^i}(X_n^i) - \mathbf{P}_i m_{\theta_n^i}(X_{n-1}^i)$ such that there exists a constant $C$,

$$
\begin{aligned}
\mathbb{E}\left[\left\|e_{n+1}^i\right\|^{2+\tau}\big|\mathcal{F}_n\right] &\leq C\mathbb{E}\left[\left\|m_{\theta_n^i}(X_n^i)\right\|^{2+\tau} + \left\|\mathbf{P}_i m_{\theta_n^i}(X_{n-1}^i)\right\|^{2+\tau}\Big|\mathcal{F}_n\right] \\
&= C\sum_{Y\in\mathcal{X}^i}\mathbf{P}_i(X_{n-1}^i,Y)\left\|m_{\theta_n^i}(Y)\right\|^{2+\tau} + C\left\|\mathbf{P}_i m_{\theta_n^i}(X_{n-1}^i)\right\|^{2+\tau}.
\end{aligned}
\tag{67}
$$

Since $\|m_{\theta_n^i}(Y)\| < \infty$ almost surely by Assumption 2.4 and $\mathcal{X}^i$ is a finite state space, at all time $n$, we have

$$
\sum_{Y\in\mathcal{X}^i}\mathbf{P}_i(X_{n-1}^i,Y)\left\|m_{\theta_n^i}(Y)\right\|^{2+\tau} < \infty \quad a.s.
\tag{68}
$$

and there exists another constant $C'$ such that by definition of $\mathbf{P}_i m_{\theta_n^i}(X_{n-1}^i)$, we have

$$
\left\|\mathbf{P}_i m_{\theta_n^i}(X_{n-1}^i)\right\|^{2+\tau} \leq C'\sum_{Y\in\mathcal{X}^i}\mathbf{P}_i(X_{n-1}^i,Y)\left\|m_{\theta_n^i}(Y)\right\|^{2+\tau} < \infty \quad a.s.
\tag{69}
$$

Therefore, $\mathbb{E}[\|e_{n+1}^i\|^{2+\tau}|\mathcal{F}_n] < \infty$ a.s. for all $n$ and C4.(a) is satisfied.

We now turn to C4.(b). Note that for any $i \neq j$, we have $\mathbb{E}[e_{n+1}^i(e_{n+1}^j)^T|\mathcal{F}_n] = \mathbb{E}[e_{n+1}^i|\mathcal{F}_n] \cdot \mathbb{E}[(e_{n+1}^j)^T|\mathcal{F}_n] = 0$ due to the independence between agent $i$ and $j$, and $\mathbb{E}[e_{n+1}^i|\mathcal{F}_n] = 0$. Then, we have

$$
\mathbb{E}\left[\left(\frac{1}{N}\sum_{i=1}^N e_{n+1}^i\right)\left(\frac{1}{N}\sum_{i=1}^N e_{n+1}^i\right)^T\bigg|\mathcal{F}_n\right] = \frac{1}{N^2}\sum_{i=1}^N\mathbb{E}\left[e_{n+1}^i(e_{n+1}^i)^T\big|\mathcal{F}_n\right].
\tag{70}
$$

The analysis of $\mathbb{E}[e_{n+1}^i(e_{n+1}^i)^T|\mathcal{F}_n]$ is inspired by Section 4 Fort (2015) and Section 4.3.3 Delyon (2000), where they constructed another Poisson equation to further decompose the noise terms therein.[8] Here, expanding $\mathbb{E}[e_{n+1}^i(e_{n+1}^i)^T|\mathcal{F}_n]$ gives

$$
\begin{aligned}
\mathbb{E}\left[e_{n+1}^i(e_{n+1}^i)^T\big|\mathcal{F}_n\right] =&\mathbb{E}[m_{\theta_n^i}(X_n^i)m_{\theta_n^i}(X_n^i)^T|\mathcal{F}_n] + \mathbf{P}_i m_{\theta_n^i}(X_{n-1}^i)\left(\mathbf{P}_i m_{\theta_n^i}(X_{n-1}^i)\right)^T \\
&-\mathbb{E}[m_{\theta_n^i}(X_n^i)|\mathcal{F}_n]\left(\mathbf{P}_i m_{\theta_n^i}(X_{n-1}^i)\right)^T - \mathbf{P}_i m_{\theta_n^i}(X_{n-1}^i)\mathbb{E}[m_{\theta_n^i}(X_n^i)^T|\mathcal{F}_n] \\
=&\sum_{y\in\mathcal{X}_i}\mathbf{P}_i(X_{n-1},y)m_{\theta_n^i}(y)m_{\theta_n^i}(y)^T - \mathbf{P}_i m_{\theta_n^i}(X_{n-1}^i)\left(\mathbf{P}_i m_{\theta_n^i}(X_{n-1}^i)\right)^T.
\end{aligned}
\tag{71}
$$

Denote by

$$
G_i(\theta^i,x) \triangleq \sum_{y\in\mathcal{X}_i}\mathbf{P}_i(x,y)m_{\theta^i}(y)m_{\theta^i}(y)^T - \mathbf{P}_i m_{\theta^i}(x)\left(\mathbf{P}_i m_{\theta^i}(x)\right)^T,
\tag{72}
$$

and let its expectation w.r.t the stationary distribution $\boldsymbol{\pi}_i$ be $g_i(\theta^i) \triangleq \mathbb{E}_{x\sim\boldsymbol{\pi}_i}[G_i(\theta^i,x)]$, we can construct another Poisson equation, i.e.,

$$
\begin{aligned}
&\mathbb{E}\left[e_{n+1}^i(e_{n+1}^i)^T\big|\mathcal{F}_n\right] - \sum_{X_n^i\in\mathcal{X}_i}\boldsymbol{\pi}(X_n^i)\mathbb{E}\left[e_{n+1}^i(e_{n+1}^i)^T\big|\mathcal{F}_n\right] \\
=&G_i(\theta_n^i,X_{n-1}^i) - g_i(\theta_n^i) \\
=&\varphi_{\theta_n^i}^i(X_{n-1}^i) - \mathbf{P}_i\varphi_{\theta_n^i}^i(X_{n-1}^i),
\end{aligned}
\tag{73}
$$

---

[8]However, we note that Fort (2015); Delyon (2000) considered the Lipschitz continuity of function $F_{\theta^i}^i(x)$ defined in (74) as an assumption instead of a conclusion, where we give a detailed proof for this. We also obtain matrix $\mathbf{U}_i$ in an explicit form, which coincides with the definition of *asymptotic covariance matrix* and was not simplified in Fort (2015). The discussion on the improvement of $\mathbf{U}_i$ is outlined in Section 3.2, which was not the focus of Fort (2015); Delyon (2000) and was not covered therein.

for some matrix-valued function $\varphi^i : \mathbb{R}^d \times \mathcal{X}_i \to \mathbb{R}^{d \times d}$. Following the similar steps shown in (48) - (60), we can obtain the closed-form expression

$$\varphi_{\theta^i}^i(x) = \sum_{y \in \mathcal{X}_i} \left(\mathbf{I} - \mathbf{P}_i + \mathbf{1}(\boldsymbol{\pi}_i)^T\right)^{-1}(x,y)G_i(\theta^i, x) - g_i(\theta^i). \tag{74}$$

Then, we can decompose (71) into

$$G_i(\theta_n^i, X_{n-1}^i) = \underbrace{g_i(\theta^*)}_{\mathbf{U}_i} + \underbrace{g_i(\theta_n^i) - g_i(\theta^*)}_{\mathbf{D}_{i,n}^{(1)}} + \underbrace{\varphi_{\theta_n^i}^i(X_n^i) - \mathbf{P}_i\varphi_{\theta_n^i}^i(X_{n-1}^i)}_{\mathbf{D}_{i,n}^{(2,a)}} + \underbrace{\varphi_{\theta_n^i}^i(X_{n-1}^i) - \varphi_{\theta_n^i}^i(X_n^i)}_{\mathbf{D}_{i,n}^{(2,b)}}.$$
$$\tag{75}$$

Let $\mathbf{U} \triangleq \frac{1}{N^2}\sum_{i=1}^N \mathbf{U}_i$, $\mathbf{D}_n^{(1)} \triangleq \frac{1}{N^2}\sum_{i=1}^N \mathbf{D}_{1,n}^{(1)}$, $\mathbf{D}_n^{(2,a)} \triangleq \frac{1}{N^2}\sum_{i=1}^N \mathbf{D}_{i,n}^{(2,a)}$, and $\mathbf{D}_n^{(2,b)} \triangleq \frac{1}{N^2}\sum_{i=1}^N \mathbf{D}_{i,n}^{(2,b)}$, we want to prove that $\mathbf{D}_n^{(1)}$ satisfies the first condition in C4, and $\mathbf{D}_n^{(2,a)}, \mathbf{D}_n^{(2,b)}$ meet the second condition in C4.

We now show that for all $i$, $G_i(\theta^i, x)$ is Lipschitz continuous in $\theta^i \in \Omega$ for some compact subset $\Omega \subset \mathbb{R}^d$. For any $x \in \mathcal{X}_i$ and $\theta_1^i, \theta_2^i \in \Omega$, we can get

$$\begin{aligned}
&\|m_{\theta_1^i}(x)m_{\theta_1^i}(x)^T - m_{\theta_2^i}(x)m_{\theta_2^i}(x)^T\| \\
=&\|m_{\theta_1^i}(x)(m_{\theta_1^i}(x) - m_{\theta_2^i}(x))^T - (m_{\theta_1^i}(x) - m_{\theta_2^i}(x))m_{\theta_2^i}(x)^T\| \\
\leq&\|m_{\theta_1^i}(x) - m_{\theta_2^i}(x)\|(\|m_{\theta_1^i}(x)\| + \|m_{\theta_2^i}(x)\|) \\
\leq&C\|\theta_1^i - \theta_2^i\|,
\end{aligned} \tag{76}$$

for some constant $C$, where the last inequality comes from $\|m_{\theta^i}(x)\| < \infty$ since $\theta_1^i \in \Omega$ and the Lipschitz continuous function $m_{\theta^i}(x)$. Similarly, we can get $\|\mathbf{P}_i m_{\theta_1^i}(x) - \mathbf{P}_i m_{\theta_2^i}(x)\| \leq C\|\theta_1^i - \theta_2^i\|$. Therefore, $G_i(\theta^i, x)$ and $g_i(\theta^i)$ are Lipschitz continuous in $\theta^i \in \Omega$ for any $x \in \mathcal{X}_i$.

For the sequence $\{\mathbf{D}_{i,n}^{(1)}\}_{n \geq 0}$, by applying Theorem 3.2 and conditioned on $\lim_{n\to\infty} \theta_n = \theta^*$ for an optimal point $\theta^* \in \mathcal{L}$, we have $\lim_{n\to\infty} \|g_i(\theta_n^i) - g_i(\theta^*)\| \leq \lim_{n\to\infty} C\|\theta_n^i - \theta^*\| = 0$. This implies $\mathbf{D}_{i,n}^{(1)} \to 0$ for every $i \in [N]$ and thus $\mathbf{D}_n^{(1)} \to 0$ as $n \to \infty$ almost surely, which satisfies the first condition in (63).

For the Martingale difference sequence $\{\mathbf{D}_{i,n}^{(2,a)}\}_{n \geq 0}$, we use Burkholder inequality (e.g., Theorem 2.10 Hall et al. (2014), Davis (1970)) such that for $p \geq 1$ and some constant $C_p$,

$$\mathbb{E}\left[\left\|\sum_{i=1}^n \mathbf{D}_{i,n}^{(2,a)}\right\|^p\right] \leq C_p \mathbb{E}\left[\left(\sum_{i=1}^n \left\|\mathbf{D}_{i,n}^{(2,a)}\right\|^2\right)^{p/2}\right]. \tag{77}$$

By the definition (72) and Assumption 2.4, for a sample path, $\sup_n \|G_i(\theta_n^i, x)\| < \infty$ for any $x \in \mathcal{X}_i$, as well as $\sup_n \|g_i(\theta_n^i)\| < \infty$, which leads to $\sup_n \|\varphi_{\theta_n^i}^i(x)\| < \infty$ for any $x \in \mathcal{X}_i$ because of (74). Then, we have $\sup_n \|\mathbf{D}_{i,n}^{(2,a)}\| \leq C < \infty$ for the path-dependent constant $C$. Taking $p = 1$ and we have

$$\lim_{n\to\infty} \gamma_n C_p \sqrt{\sum_{i=1}^n \left\|\mathbf{D}_{i,n}^{(2,a)}\right\|^2} \leq \lim_{n\to\infty} C_p C \gamma_n \sqrt{n} = 0 \quad a.s. \tag{78}$$

Thus, Lebesgue dominated convergence theorem gives

$$\lim_{n\to\infty} \gamma_n C_p \mathbb{E}\left[\sqrt{\sum_{i=1}^n \|\mathbf{D}_{i,n}^{(2,a)}\|^2}\right] = \mathbb{E}\left[\lim_{n\to\infty} \gamma_n C_p \sqrt{\sum_{i=1}^n \|\mathbf{D}_{i,n}^{(2,a)}\|^2}\right] = 0$$

and we have $\lim_{n\to\infty} \gamma_n \mathbb{E}[\|\sum_{i=1}^n \mathbf{D}_{i,n}^{(2,a)}\|] = 0$.

For the sequence $\{\mathbf{D}_{i,n}^{(2,b)}\}_{n\geq 0}$, we have

$$\sum_{k=1}^{n} \mathbf{D}_{i,k}^{(2,b)} = \sum_{k=1}^{n} \left( \varphi_{\theta_k^i}^i(X_{k-1}^i) - \varphi_{\theta_{k-1}^i}^i(X_{k-1}^i) \right) + \varphi_{\theta_0^i}^i(X_0^i) - \varphi_{\theta_n^i}^i(X_n^i)$$

$$= \sum_{k=1}^{n} \left( \varphi_{\theta_k^i}^i(X_{k-1}^i) - \varphi_{\theta_k}^i(X_{k-1}^i) + \varphi_{\theta_k}^i(X_{k-1}^i) - \varphi_{\theta_{k-1}}^i(X_{k-1}^i) + \varphi_{\theta_{k-1}}^i(X_{k-1}^i) - \varphi_{\theta_{k-1}^i}^i(X_{k-1}^i) \right)$$
$$+ \varphi_{\theta_0^i}^i(X_0^i) - \varphi_{\theta_n^i}^i(X_n^i). \tag{79}$$

Since $G_i(\theta^i, x)$ and $g_i(\theta^i)$ are Lipschitz continuous in $\theta^i \in \Omega$, $\varphi_{\theta^i}^i(x)$ is also Lipschitz continuous in $\theta^i \in \Omega$ and is bounded. We have

$$\left\| \sum_{k=1}^{n} \mathbf{D}_{i,k}^{(2,b)} \right\| \leq \left\| \sum_{k=1}^{n} \varphi_{\theta_k^i}^i(X_{k-1}^i) - \varphi_{\theta_{k-1}^i}^i(X_{k-1}^i) \right\| + \left\| \varphi_{\theta_0^i}^i(X_0^i) \right\| + \left\| \varphi_{\theta_n^i}^i(X_n^i) \right\|$$

$$\leq \left\| \sum_{k=1}^{n} \varphi_{\theta_k^i}^i(X_{k-1}^i) - \varphi_{\theta_{k-1}^i}^i(X_{k-1}^i) \right\| + D_1 \tag{80}$$

$$\leq \sum_{k=1}^{n} D_2 D_\Omega \gamma_k + D_1$$

where $\|\varphi_{\theta_0^i}^i(X_0^i)\| + \|\varphi_{\theta_n^i}^i(X_n^i)\| \leq D_1$ for a given sample path, $D_2$ is the Lipschitz constant of $\varphi_{\theta^i}^i(x)$, and $\|\nabla F_i(x^i, X^i)\| \leq D_\Omega$ for any $x^i \in \Omega$ and $X^i \in \mathcal{X}^i$. Then,

$$\gamma_n \left\| \sum_{k=1}^{n} \mathbf{D}_{i,k}^{(2,b)} \right\| \leq D_2 D_\Omega \gamma_n \sum_{k=1}^{n} \gamma_k + \gamma_n D_1 \to 0 \quad \text{as } n \to \infty \tag{81}$$

because $\gamma_n \sum_{k=1}^{n} \gamma_k = O(n^{1-2a})$ by assumption 2.3. Therefore, the second condition of C4 is satisfied.

### E.3 Analysis of C5

We now analyze condition C5. The decreasing rate of each term in (57) has been proved in Appendix D. Specifically, by assumption 2.4, there exists a compact subset for a given sample path, and

- we have shown that $\left\| r_n^{(A)} \right\| = O(\eta_n)$ a.s., which implies $\left\| r_n^{(A)} \right\| = o(\sqrt{\gamma_n})$ a.s.

- For $\frac{1}{N} \sum_{i=1}^{N} \xi_n^i$, in the case of increasing communication interval, $\frac{1}{N} \sum_{i=1}^{N} \xi_n^i = O(\gamma_n + \eta_n)$, by Assumption 2.3-ii), we know $(\gamma_n + \eta_n)/\sqrt{\gamma_n} = \sqrt{\gamma_n} + \sqrt{\gamma_n} K_{\tau_n}^{L+1} = o(1)$ such that $\|\frac{1}{N} \sum_{i=1}^{N} \xi_n^i\| = o(\sqrt{\gamma_n})$ almost surely. On the other hand, in the case of bounded communication interval, $\frac{1}{N} \sum_{i=1}^{N} \xi_n^i = O(\gamma_n)$ such that $\|\frac{1}{N} \sum_{i=1}^{N} \xi_n^i\| = o(\sqrt{\gamma_n})$ a.s.

- Since $\sup_n \|\nu_n^i\| < \infty$ almost surely, we have $\sup_p \|\frac{1}{N} \sum_{i=1}^{N} \sum_{k=0}^{p} (\nu_k^i - \nu_{k+1}^i)\| = \sup_p \|\frac{1}{N} \sum_{i=1}^{N} (\nu_0^i - \nu_{p+1}^i)\| < \infty$ almost surely. Then, $\sqrt{\gamma_p} \|\frac{1}{N} \sum_{i=1}^{N} \sum_{k=0}^{p} (\nu_k^i - \nu_{k+1}^i)\| = O(\sqrt{\gamma_p})$ leads to $\sqrt{\gamma_p} \|\frac{1}{N} \sum_{i=1}^{N} \sum_{k=0}^{p} (\nu_k^i - \nu_{k+1}^i)\| = o(1)$ a.s.

Let $r_n^{(1)} \triangleq r_n^{(A)} + \frac{1}{N} \sum_{i=1}^{N} \xi_n^i$ and $r_n^{(2)} \triangleq \frac{1}{N} \sum_{i=1}^{N} (\nu_k^i - \nu_{k+1}^i)$. From above, we can see that C5 in Theorem E.1 is satisfied and we show that all the conditions in Theorem E.1 have been satisfied.

### E.4 Closed Form of Limitimg Covariance Matrix

Lastly, we need to analyze the closed-form expression of $\mathbf{U}$ as in C4 (b) of Theorem E.1. Recall that $\mathbf{U} = \frac{1}{N^2} \sum_{i=1}^{N} \mathbf{U}_i$ and $\mathbf{U}_i = g_i(\theta^*)$ in (75). We now give the exact form of function $g_i(\theta^*)$ as

follows:

$$
\begin{aligned}
g_i(\theta^*) &= \sum_{x \in \mathcal{X}_i} \boldsymbol{\pi}_i(x) \left[ m_{\theta^*}(x) m_{\theta^*}(x)^T - \left( \sum_{y \in \mathcal{X}_i} \mathbf{P}_i(x,y) m_{\theta^*}(y) \right) \left( \sum_{y \in \mathcal{X}_i} \mathbf{P}_i(x,y) m_{\theta^*}(y) \right)^T \right] \\
&= \mathbb{E}\left[ \left( \sum_{s=0}^{\infty} [\nabla F_i(\theta^*, X_s) - \nabla f_i(\theta^*)] \right) \left( \sum_{s=0}^{\infty} [\nabla F_i(\theta^*, X_s) - \nabla f_i(\theta^*)] \right)^T \right] \\
&\quad - \mathbb{E}\left[ \left( \sum_{s=1}^{\infty} [\nabla F_i(\theta^*, X_s) - \nabla f_i(\theta^*)] \right) \left( \sum_{s=1}^{\infty} [\nabla F_i(\theta^*, X_s) - \nabla f_i(\theta^*)] \right)^T \right] \\
&= \mathbb{E}\left[ \left( \nabla F_i(\theta^*, X_0^i) - \nabla f_i(\theta^*) \right) \left( \nabla F_i(\theta^*, X_0^i) - \nabla f_i(\theta^*) \right)^T \right] \\
&\quad + \mathbb{E}\left[ \left( \nabla F_i(\theta^*, X_0^i) - \nabla f_i(\theta^*) \right) \left( \sum_{s=1}^{\infty} [\nabla F_i(\theta^*, X_s) - \nabla f_i(\theta^*)] \right)^T \right] \\
&\quad + \mathbb{E}\left[ \left( \sum_{s=1}^{\infty} [\nabla F_i(\theta^*, X_s) - \nabla f_i(\theta^*)] \right) \left( \nabla F_i(\theta^*, X_0^i) - \nabla f_i(\theta^*) \right)^T \right] \\
&= \mathrm{Cov}(\nabla F_i(\theta^*, X_0), \nabla F_i(\theta^*, X_0)) \\
&\quad + \sum_{s=1}^{\infty} \left[ \mathrm{Cov}(\nabla F_i(\theta^*, X_0), \nabla F_i(\theta^*, X_s)) + \mathrm{Cov}(\nabla F_i(\theta^*, X_s), \nabla F_i(\theta^*, X_0)) \right], \\
&= \boldsymbol{\Sigma}(\nabla F(\theta^*, \cdot)).
\end{aligned}
\tag{82}
$$

where the second equality comes from the recursive form of $m_{\theta^i}(x)$ in (60), and that the process $\{X_n\}_{n \geq 0}$ is in its stationary regime, i.e., $X_0 \sim \boldsymbol{\pi}_i$ from the beginning. The last equality comes from rewriting $\mathrm{Cov}(\nabla F_i(\theta^*, X_i), \nabla F_i(\theta^*, X_j))$ in a matrix form. Note that $g_i(\theta^*)$ is exactly the asymptotic covariance matrix of the underlying Markov chain $\{X_n^i\}_{n \geq 0}$ associated with the test function $\nabla F_i(\theta^*, \cdot)$. By utilizing the following lemma, we can obtain the explicit form of $\mathbf{V}$ as defined in (66).

**Lemma E.2.** *If all the eigenvalues of matrix $\mathbf{M}$ have negative real part, then for every positive semi-definite matrix $\mathbf{U}$ there exists a unique positive semi-definite matrix $\mathbf{V}$ satisfying $\mathbf{U} + \mathbf{M}\mathbf{V} + \mathbf{V}\mathbf{M}^T = \mathbf{0}$. The explicit solution $\mathbf{V}$ is given as*

$$
\mathbf{V} = \int_0^{\infty} e^{\mathbf{M}t} \mathbf{U} e^{(\mathbf{M}^T)t} dt.
\tag{83}
$$

*Proof.* Most of the following steps are from the proof of Theorem 3.16 Chellaboina & Haddad (2008). However, it requires positive definite matrix $\mathbf{U}$, which is not needed in this paper. Therefore, we attach the proof of Lemma E.2 with relaxed condition on matrix $\mathbf{U}$ (e.g., being positive semi-definite instead of positive definite) for self-contained purpose.

We first show that $\mathbf{V}$ is the solution to the equation $\mathbf{U} + \mathbf{M}\mathbf{V} + \mathbf{V}\mathbf{M}^T = \mathbf{0}$. Note that

$$
\begin{aligned}
\mathbf{M}\mathbf{V} + \mathbf{V}\mathbf{M}^T &= \int_0^{\infty} \mathbf{M} e^{\mathbf{M}t} \mathbf{U} e^{(\mathbf{M}^T)t} dt + \int_0^{\infty} e^{\mathbf{M}t} \mathbf{U} e^{(\mathbf{M}^T)t} \mathbf{M}^T dt \\
&= \int_0^{\infty} \frac{d}{dt} \left( e^{\mathbf{M}t} \mathbf{U} e^{\mathbf{M}^T t} \right) dt \\
&= e^{\mathbf{M}t} \mathbf{U} e^{\mathbf{M}^T t} \Big|_0^{\infty} \\
&= -\mathbf{U},
\end{aligned}
\tag{84}
$$

which implies that $\mathbf{V}$ in the form of (83) is a solution to the equation $\mathbf{U} + \mathbf{M}\mathbf{V} + \mathbf{V}\mathbf{M}^T = \mathbf{0}$. Since the integrand in (83) involves a sum of terms of the form $e^{\lambda_i t}$, where $\{\lambda_i\}_{i \in [d]}$ are the eigenvalues of matrix $\mathbf{M}$ and $Re(\lambda_i) < 0$, the integral in (83) is well-defined.

Next, we show the uniqueness of $\mathbf{V}$. Assume there exist two solutions $\mathbf{V}_1$ and $\mathbf{V}_2$ and $\mathbf{V}_1 \neq \mathbf{V}_2$. Then,

$$\mathbf{MV}_1 + \mathbf{V}_1\mathbf{M}^T + \mathbf{U} = \mathbf{0}, \tag{85a}$$

$$\mathbf{MV}_2 + \mathbf{V}_2\mathbf{M}^T + \mathbf{U} = \mathbf{0}. \tag{85b}$$

Subtracting (85b) from (85a) yields

$$\mathbf{M}(\mathbf{V}_1 - \mathbf{V}_2) + (\mathbf{V}_1 - \mathbf{V}_2)\mathbf{M}^T = \mathbf{0}. \tag{86}$$

Then,

$$\mathbf{0} = e^{\mathbf{M}t}\left[\mathbf{M}(\mathbf{V}_1 - \mathbf{V}_2) + (\mathbf{V}_1 - \mathbf{V}_2)\mathbf{M}^T\right] = \frac{d}{dt}\left(e^{\mathbf{M}t}(\mathbf{V}_1 - \mathbf{V}_2)e^{\mathbf{M}^T}t\right)dt. \tag{87}$$

Taking integral of (87) on both sides gives

$$\mathbf{0} = \int_0^\infty \frac{d}{dt}\left(e^{\mathbf{M}t}(\mathbf{V}_1 - \mathbf{V}_2)e^{\mathbf{M}^T t}\right)dt = e^{\mathbf{M}t}(\mathbf{V}_1 - \mathbf{V}_2)e^{\mathbf{M}^T t}\Big|_0^\infty = \mathbf{V}_2 - \mathbf{V}_1. \tag{88}$$

This contradicts to the assumption $\mathbf{V}_1 \neq \mathbf{V}_2$, and hence, $\mathbf{V}$ in (83) is the unique solution to the equation $\mathbf{U} + \mathbf{MV} + \mathbf{VM}^T = \mathbf{0}$. $\qquad\square$

## E.5 CLT of Polyak-Ruppert Averaging

We now consider the CLT result of Polyak-Ruppert averaging $\bar{\theta}_n = \sum_{k=0}^{n-1}\theta_k$. The steps follow similar way by verifying that the conditions in the related CLT of Polyak-Ruppert averaging for the stochastic approximation are satisfied. The additional assumption is given below.

C6. For the sequence $\{r_n\}$ in (44), $n^{-1/2}\sum_{k=0}^n r_k^{(1)} \to 0$ with probability 1.

Then, the CLT of Polyak-Ruppert averaging is as follows.

**Theorem E.3** (Theorem 3.2 of Fort (2015)). *Consider the iteration* (44)*, assume C1, C3, C4, C5 in Theorem E.1 are satisfied. Moreover, assume C6 is satisfied. Then, with step size $\gamma_n \sim \gamma_\star/n^a$ for $a \in (0.5, 1)$, we have*

$$\sqrt{n}(\bar{\theta}_n - \theta^*) \xrightarrow[n\to\infty]{dist.} \mathcal{N}(0, \mathbf{V}'), \tag{89}$$

*where $\mathbf{V}' = \mathbf{H}^{-1}\mathbf{UH}^{-T}$.*

Discussion about C1 and C3 can be found in Section E.1. Condition C4 has been analyzed in Section E.2 and condition C5 has been examined in Section E.3. The only condition left to analyze is C6, which is based on the results obtained in Section E.3. In view of (57), $r_n^{(1)} = r_n^{(A)} + \frac{1}{N}\sum_{i=1}^N \xi_{n+1}^i$, so C6 is equivalent to

$$n^{-1/2}\sum_{k=1}^n\left[r_k^{(A)} + \frac{1}{N}\sum_{i=1}^N\left(\xi_{k+1}^i\right)\right] \to 0 \quad w.p.1. \tag{90}$$

In Section E.3, we have shown that $\left\|r_n^{(A)}\right\| = O(\eta_n)$, $\frac{1}{N}\sum_{i=1}^N \xi_n^i = O(\gamma_n)$. Note that by Assumption 2.3, we consider bounded communication interval for step size $\gamma_n \sim \gamma_\star/n^a$ for $a \in (0.5, 1)$, and hence, $\eta_n = O(\gamma_n)$ such that $\left\|r_n^{(A)}\right\| = O(\gamma_n)$. We then know that

$$\sum_{k=1}^n\left\|r_n^{(A)}\right\| = O(n^{1-a}), \quad \sum_{k=1}^n\|\frac{1}{N}\sum_{i=1}^N\xi_n^i\| = O(n^{1-a}), \tag{91}$$

such that

$$n^{-1/2}\sum_{k=1}^n\left\|r_k^{(A)} + \frac{1}{N}\sum_{i=1}^N\left(\xi_{k+1}^i\right)\right\| = O(n^{1/2-a}) = o(1), \tag{92}$$

which proved (90) and C6 is verified. Therefore, Theorem E.3 is proved under our Assumptions 2.1 - 2.5.

# F DISCUSSION ON THE COMPARISON OF THEOREM 3.3 TO THE CLT RESULT IN LI ET AL. (2022)

As a byproduct of our Theorem 3.3, we have the following corollary.

**Corollary F.1.** *Under Assumptions 2.1 - 2.5, for the sub-sequence $\{n_l\}_{l \geq 0}$ where $K_l = K$ for all $l$, we have*

$$\frac{1}{\sqrt{n_l}} \sum_{k=1}^{l} (\bar{\theta}_{n_k} - \theta^*) \xrightarrow[l \to \infty]{dist.} \mathcal{N}(0, \mathbf{V}') \tag{93}$$

*Proof.* Since $K_l = K$ for all $l$, we have $n_l = Kl$. There is an existing result showing the CLT result of the partial sum of a sub-sequence (after normalization) has the same normal distribution as the partial sum of the original sequence.

**Theorem F.2** (Theorem 14.4 of Billingsley (2013)). *Given a sequence of random variable $\theta_1, \theta_2, \cdots$ with partial sum $S_n \triangleq \sum_{k=1}^{n} \theta_k$ such that $\frac{1}{\sqrt{n}} S_n \xrightarrow[n \to \infty]{dist.} \mathcal{N}(0, \mathbf{V})$. Let $n_l$ be some positive random variable taking integer value such that $\theta_{n_l}$ is on the same space as $\theta_n$. In addition, for some sequence $\{b_l\}_{l \geq 0}$ going to infinity, $n_l/b_l \to c$ for a positive constant $c$. Then, $\frac{1}{\sqrt{n_l}} S_{n_l} \xrightarrow[l \to \infty]{dist.} \mathcal{N}(0, \mathbf{V})$.*

Applying Theorem F.2 along with our Theorem 3.3, we have $\frac{1}{\sqrt{n_l}} \sum_{k=1}^{l} (\bar{\theta}_{n_k} - \theta^*) \xrightarrow[l \to \infty]{dist.} \mathcal{N}(0, \mathbf{V}')$. $\square$

Recently, Li et al. (2022) studied the CLT result under the L-SGD-FC algorithm with *i.i.d* sampling (with slightly different setting of the step size). We are able recover Theorem 3.1 of Li et al. (2022) under the constant communication interval while adjusting their step size to make a fair comparison. We state their algorithm below for self-contained purpose. During each communication interval $n \in (n_l, n_{l+1}]$,

$$\theta_{n+1}^i = \begin{cases} \theta_n^i - \gamma_l \nabla F_i(\theta_n^i, X_n^i) & \text{if } n \in (n_l, n_{l+1}), \\ \frac{1}{N} \sum_{i=1}^{N} (\theta_n^i - \gamma_l \nabla F_i(\theta_n^i, X_n^i)) & \text{if } n = n_{l+1}. \end{cases} \tag{94}$$

The CLT result associated with (94) is given below.

**Theorem F.3** (Theorem 3.1 of Li et al. (2022)). *Under L-SGD-FC algorithm with i.i.d. sampling, we have*

$$\frac{\sqrt{n_l}}{l} \sum_{k=1}^{l} (\bar{\theta}_{n_k} - \theta^*) \xrightarrow[l \to \infty]{dist.} \mathcal{N}(0, \nu \mathbf{V}'), \tag{95}$$

*where $\nu \triangleq \lim_{l \to \infty} \frac{1}{l^2} (\sum_{k=1}^{l} K_l)(\sum_{k=1}^{l} K_l^{-1})$.*

Note that $\nu = 1$ for constant $K$. We can rewrite (95) as

$$\frac{\sqrt{n_l}}{l} \sum_{k=1}^{l} (\bar{\theta}_{n_k} - \theta^*) = \frac{\sqrt{n_l}}{\sqrt{l}} \frac{1}{\sqrt{l}} \sum_{k=1}^{l} (\bar{\theta}_{n_k} - \theta^*) = \sqrt{K} \frac{1}{\sqrt{l}} \sum_{k=1}^{l} (\bar{\theta}_{n_k} - \theta^*) \tag{96}$$

such that

$$\frac{1}{\sqrt{l}} \sum_{k=1}^{l} (\bar{\theta}_{n_k} - \theta^*) \xrightarrow[l \to \infty]{dist.} \mathcal{N}(0, \frac{1}{K} \mathbf{V}'). \tag{97}$$

Note that the step size in (94) keeps unchanged during each communication interval, while the step size in our GD-SGD itereates keeps decreasing even in the same communication interval. This makes our step size decreasing faster than theirs. To make a fair comparison, we only choose a sub-sequence $\{n_{Kl}\}_{l \geq 0}$ in (97) such that it is 'equivalent' to see that our step sizes become the same at each aggregation step. In this case, we again use Theorem F.2 to obtain

$$\frac{1}{\sqrt{Kl}} \sum_{s=1}^{l} (\bar{\theta}_{n_{Ks}} - \theta^*) \xrightarrow[l \to \infty]{dist.} \mathcal{N}(0, \frac{1}{K} \mathbf{V}'), \tag{98}$$

such that

$$\frac{1}{\sqrt{l}} \sum_{s=1}^{l} (\bar{\theta}_{n_{Ks}} - \theta^*) = \sqrt{K} \frac{1}{\sqrt{Kl}} \sum_{s=1}^{l} (\bar{\theta}_{n_{Ks}} - \theta^*) \xrightarrow[l\to\infty]{dist.} \mathcal{N}(0, \mathbf{V}'). \tag{99}$$

Therefore, our Corollary F.1 also recovers Theorem 3.1 of Li et al. (2022) under the constant communication interval $K$ with more general communication patterns $\mathbf{W}$ and *Markovian* sampling.

## G  ADDITIONAL SIMULATIONS OF GD-SGD ALGORITHM

In this part, we test a non-convex objective function used in Allen-Zhu & Yuan (2016); Gower et al. (2019); Hu et al. (2022) as follows:

$$f(\theta) = \frac{1}{NB} \sum_{i=1}^{N} \sum_{j=1}^{B} \theta^T (\mathbf{x}_{i,j} \mathbf{x}_{i,j}^T + \mathbf{D}_{i,j}) \theta + \mathbf{b}^T \theta, \tag{100}$$

where $\mathbf{x}_{i,j}$ is the data point in CIFAR-10 dataset, $\mathbf{b}$ is a shared vector among all the agents where each entry is randomly generated in the range $(0, 1)$, and $\mathbf{D}_{i,j}$ is a diagonal matrix such that $\sum_{i,j} (\mathbf{x}_{i,j} \mathbf{x}_{i,j}^T + \mathbf{D}_{i,j})$ is invertible and has at least one negative eigenvalue in order to make objective function $f(\cdot)$ non-convex. We know that (100) has a unique saddle point at

$$\theta^* = \left( \frac{1}{NB} \sum_{i=1}^{N} \sum_{j=1}^{B} \mathbf{x}_{i,j} \mathbf{x}_{i,j}^T + \mathbf{D}_{i,j} \right)^{-1} \mathbf{b}. \tag{101}$$

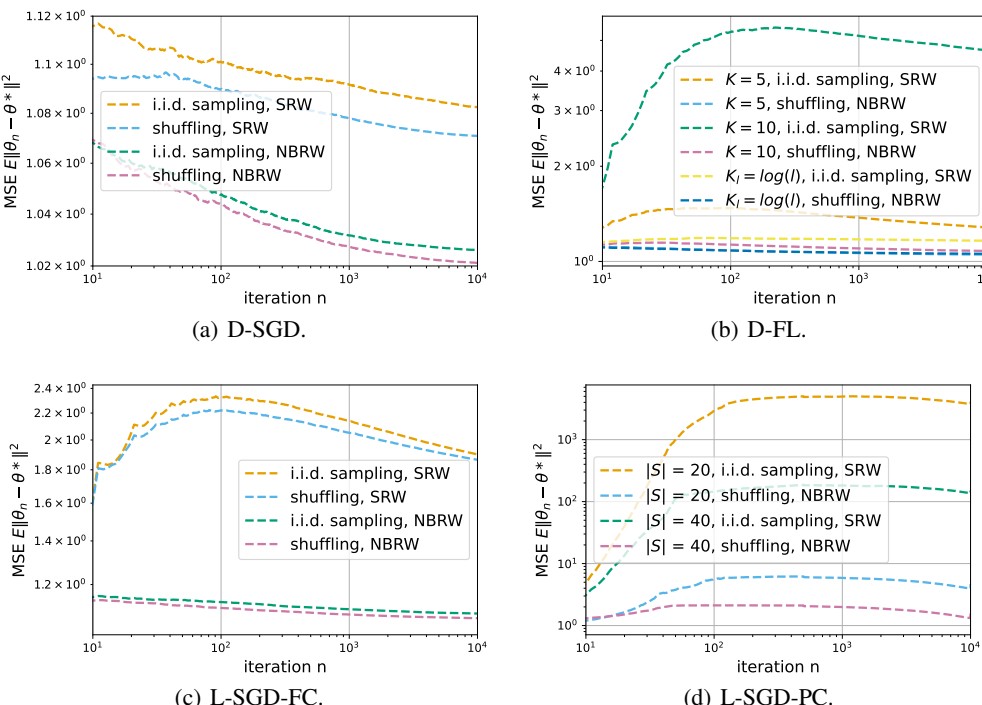

Figure 3: Simulation results for objective function (100). (a) D-SGD with four combinations of sampling strategies. (b) D-FL with different communication intervals $K = 5, 10$ and $K_l = \log(l)$. (c) L-SGD-FC with $K = 10$ and four combinations of sampling strategies. (d) L-SGD-PC with without-replacement client sampling method and the client set size $|\mathcal{S}| = 20, 40$.

Note that Figure 3 shows that the GD-SGD iterate is yet to enter the asymptotic regime and the communication pattern still affects the MSE. All four subplots imply that more efficient sampling

strategy leads to smaller MSE most of the time, e.g., single shuffling and NBRW outperform *i.i.d* sampling and SRW, even if only partial agents improve their strategies, which share the same trend as in Section 4. Moreover, we observe that communication intervals and communication patterns do affect the convergence speed in the non-asympototic regime. Recall that our analysis in Appendix C shows the consensus error at a rate of $O(\gamma_n)$, which is faster than the typical convergence rate $O(\sqrt{\gamma_n})$ in the CLT result in Theorem 3.3, and is also consistent with previous studies Pu et al. (2020); Olshevsky (2022). In Figure 3, the consensus error, although is on the higher order term, still contributes a lot to the MSE. Specifically, Figure 3 shows that the convergence speed can be affected by large $K$, fewer client participation and different communication matrix, which is consistent with the current non-asymptotic analysis in the literature (Chen et al., 2022; Cho et al., 2022; Luo et al., 2022). For the trade-off purpose, employing increasing $K_l = \log(l)$ in Figure 3(b) still performs good since it leads to frequent aggregation in the early stage while reducing the aggregation frequency for large time, seeking the balance between convergence speed and communication cost. More importantly, we observe that employing more efficient sampling strategies, e.g., NBRW and shuffling, can greatly reduce the MSE and perform better than the baseline sampling strategies, e.g., SRW and *i.i.d* sampling. For example, even for larger communication interval $K = 10$, the scenario with NBRW and shuffling still shows faster convergence than the one with baseline sampling strategies and $K = 5$. Besides, in Figure 3(d), employing efficient sampling strategies with $|\mathcal{S}| = 20$ outperforms the baseline sampling strategies with more client participation $|\mathcal{S}| = 40$. This suggests a practical conclusion to employ efficient sampling strategies to balance the loss due to less aggregation.

