# OpenReview forum: "Impact of Agent Behavior in Distributed SGD and Federated Learning"
_ICLR.cc/2024/Conference — Submitted to ICLR 2024_

### Official Review · Reviewer_qXnB · 2023-11-01

**Soundness:** 3 good
**Presentation:** 3 good
**Contribution:** 2 fair
**Rating:** 5
**Confidence:** 4

**Summary:**

The manuscript studies the asymptotic convergence of a generalized distributed SGD method (GD-SGD) for distributed leaning problem. The authors consider various communication patterns and different sampling strategies, including iid sampling and Markovian sampling, for GD-SGD. They show the influence of sampling strategies on the limiting covariance matrix according to the definition of Loewner ordering, which is also examined in a regularized logistic regression task.

**Strengths:**

1. The authors analyze the asymptotic convergence of the D-SGD algorithm under more general communication topologies and different sampling strategies including iid sampling and Markovian sampling. The theoretical analysis seems solid.

2. The paper is well-written and easy to follow.

**Weaknesses:**

1. There have been many studies on communication topology in existing work, e.g., [Koloskova et al. (2020), Wang et al. (2021)]. Generally speaking, as long as assumption 2.5 is made, the consistency of the distributed learning algorithm can be guaranteed, so the GD-SGD algorithm designed in this paper is not novel.

2. Technically, the main proof techniques used in the paper can be found in [Li et al. (2022)] and [Hu et al. (2022)], except for the expansion of the communication patterns. Therefore, combined with the first weakness, the technical contribution of the paper is insufficient.

3. The analysis and comparison of different sampling strategies in Cor. 3.4 are trivial. The authors only give a qualitative comparison of different sampling strategies based on existing work [Hu et al. (2022)]. In fact, this simple relationship can be easily generalized in existing works with both asymptotical and non- asymptotical results. From this point of view, the contribution of this article seems to be over-claimed.

4. Logistic regression is a toy model, it is better to further consider other real-world models.

Reference:

Anastasia Koloskova, Nicolas Loizou, Sadra Boreiri, Martin Jaggi, and Sebastian Stich. A unified theory of decentralized sgd with changing topology and local updates. In International Conference on Machine Learning, pp. 5381–5393, 2020.

Wang, Jianyu, and Gauri Joshi. "Cooperative SGD: A unified framework for the design and analysis of local-update SGD algorithms." The Journal of Machine Learning Research 22.1 (2021): 9709-9758.

Xiang Li, Jiadong Liang, Xiangyu Chang, and Zhihua Zhang. Statistical estimation and online inference via local sgd. In Proceedings of Thirty Fifth Conference on Learning Theory, volume 178 of Proceedings of Machine Learning Research, pp. 1613–1661, 02–05 Jul 2022.

Jie Hu, Vishwaraj Doshi, and Do Young Eun. Efficiency ordering of stochastic gradient descent. In Advances in Neural Information Processing Systems, 2022.

**Questions:**

One of the key concern of the reviewer is on the fundamental difference in proof techniques compared to [Li et al. (2022)] and [Hu et al. (2022)]; the authors should properly address this.

Another concern of the reviewer is that the results established in this paper are in an asymptotic sense; can these results be extended to non-asymptotic ones?

---

> ### Author Response · Authors · 2023-11-19
> **Response to Reviewer qXnB (1/2)**
>
> We thank the reviewer for their detailed comments. Following are our detailed responses to the questions posed.
>
> >### (Q1). There have been many studies on communication topology in existing work, e.g., [Koloskova et al. (2020), Wang et al. (2021)]. Generally speaking, as long as assumption 2.5 is made, the consistency of the distributed learning algorithm can be guaranteed, so the GD-SGD algorithm designed in this paper is not novel.
>
> Answer: We acknowledge that Assumption 2.5 is indeed a common condition to guarantee the consistency of various distributed learning algorithms. However, the primary aim of our paper is not to introduce a novel distributed learning algorithm per se. Rather, our focus is on providing a detailed asymptotic analysis within the unified framework of *generalized* decentralized SGD (GD-SGD), by showing the consistency (almost sure convergence) as well as asymptotic normality via CLT, with which we can single out the impact of each agent on the overall performance. The essence of our contribution lies in elucidating the effects of different sampling strategies employed by individual agents in a distributed learning setting. This aspect of our research is particularly significant as it addresses nuances that are not captured by the current non-asymptotic analysis frameworks, as highlighted in Table 1 of our paper.
>
> >### (Q2). Technically, the main proof techniques used in the paper can be found in [Li et al. (2022)] and [Hu et al. (2022)], except for the expansion of the communication patterns. Therefore, combined with the first weakness, the technical contribution of the paper is insufficient.
>
> Answer: Please refer to the response to (Q1) in the ‘response to all reviewers’ part.
>
> >### (Q3). The analysis and comparison of different sampling strategies in Cor. 3.4 are trivial. The authors only give a qualitative comparison of different sampling strategies based on existing work [Hu et al. (2022)]. In fact, this simple relationship can be easily generalized in existing works with both asymptotical and non- asymptotical results. From this point of view, the contribution of this article seems to be over-claimed.
>
> Answer: Our study, indeed, uses efficient sampling strategies like non-backtracking random walks and shuffling methods as an application of our main CLT result in Theorem 3.3. However, these examples serve merely as a demonstration of the broader applicability of our theoretical findings rather than being the focal point of our study.
> The core of our contribution lies in the generality of our model and the unified results we offer within the broad framework of decentralized learning, where we uncover that the sampling strategy of each individual agent affects the overall performance while the effect of communication pattern contributes only via its leading eigenvector (stationary distribution) under the most general setup. This approach is fundamentally different from the specific settings explored in [Hu et al. (2022)]. While [Hu et al. (2022)] focus on certain types of sampling in single-agent settings, our work represents not just a straightforward extension of these concepts into a much more complex multi-agent environment with versatile communication patterns, but a substantial expansion in technical analysis in handling the consensus error among multiple agents under Markovian sampling with increasing communication interval.
> As we have highlighted in the response to (Q1) in the ‘response to all reviewers’ part, finite-time analyses, while valuable, do not differentiate the impact of each individual agent’s sampling strategy to the same extent as our asymptotic approach. They also fail to isolate long-lasting factors that significantly influence performance over extended periods. While certain communication matrices might offer short-term benefits, our numerical results in Figure 2 demonstrate that they have minimal long-term effects on the MSE with the same sampling strategy setup, which is supported by findings in [Pu et al. 2020, Olshevsky 2022]. In contrast, our new results under most general settings affirm that the sampling strategies of agents have a lasting and more substantial influence on the system's performance.
>
> >Pu, S., Olshevsky, A., & Paschalidis, I. C. Asymptotic network independence in distributed
> stochastic optimization for machine learning: Examining distributed and centralized stochastic
> gradient descent. IEEE signal processing magazine, 37(3):114–122, 2020.
> >
> >Olshevsky, A. Asymptotic network independence and step-size for a distributed subgradient
> method. Journal of Machine Learning Research, 23(69):1–32, 2022.

---

> ### Author Response · Authors · 2023-11-19
> **Response to Reviewer qXnB (2/2)**
>
> >### (Q4). Logistic regression is a toy model, it is better to further consider other real-world models.
>
> Answer: We understand the importance of demonstrating the applicability of our GD-SGD algorithm to more complex, real-world scenarios. In response to this, we would like to clarify that we have also performed the simulation with a non-convex objective function in Appendix G in our original submission. In this additional simulation, the GD-SGD algorithm has not yet reached the asymptotic regime, and the communication patterns (e.g., size of partial client sampling set, communication interval, and communication matrix) still significantly influence the MSE. Notably, the core finding from our study remained consistent: enhanced sampling strategies employed by a subset of agents resulted in faster convergence rates across almost all time periods. This result is particularly insightful as it suggests that employing effective sampling strategies at the level of individual agents can mitigate performance losses that might arise from less frequent aggregation. Such a strategy is valuable in reducing communication costs, whether among agents or between agents and a central server.
>
> >### (Q5). The results established in this paper are in an asymptotic sense; can these results be extended to non-asymptotic ones?
>
> Answer: Please see the response to (Q2) in the ‘response to all reviewers’ part.

---

> > ### Comment · Reviewer_qXnB · 2023-11-23
> > **Response to the rebuttal**
> >
> > The reviewer thank the authors's effort in the reply which have partially addressed the reviewer's concerns. The reviewer would thus maintain the score.

---

### Official Review · Reviewer_iur4 · 2023-11-07

**Soundness:** 3 good
**Presentation:** 2 fair
**Contribution:** 2 fair
**Rating:** 3
**Confidence:** 3

**Summary:**

This work revolves around distributed learning and specifically studies  the asymptotic behavior of Generalized Distributed Gradient SGD under various communication patterns and sampling strategies. The authors provide theoretical results showing asymptotic consensus convergence across clients and analyze the impact of different sampling strategies on the limiting covariance matrix. Those results provide useful insights and the generalized framework under consideration incorporates numerous results as special cases such as SGD and Distributed SGD. Experimental results on CIFAR10 further support the theoretical findings.

**Strengths:**

-This paper studies an interesting framework in distributed learning. Analyzing the Generalized Distributed SGD provides useful insights and the derived theoretical results are aligned with the results from numerous prior works (observed as special cases).

-The importance of sampling strategies for the convergence rate is being explored as well as different communication patterns in Generalized Distributed SGD.

**Weaknesses:**

-The theoretical results of this paper appear to be straightforward extensions of existing works (Morral et al., 2017; Koloskova et al., 2020; Hu et al., 2022). As a result the theoretical contribution, novelty and impact of this work appears to be marginal.

-The analysis although insightful is asymptotic in nature which somewhat diminishes the impact of the results.

-Although, there is extensive description on how the current findings are aligned with known results, the authors do not emphasize enough on the new challenges they had to overcome in order to derive their theoretical results or discuss how their work is more challenging from related works.

-The structure of the introduction could be improved curving out a related work section.

-The experimental results provided are limited to the CIFAR10 dataset.

**Questions:**

See weaknesses section.

---

> ### Author Response · Authors · 2023-11-19
> **Response to Reviewer iur4**
>
> Our sincere thanks for the in-depth review of our paper. We now answer the question posed by the reviewer.
>
> >### (Q1). The theoretical results of this paper appear to be straightforward extensions of existing works (Morral et al., 2017; Koloskova et al., 2020; Hu et al., 2022). As a result the theoretical contribution, novelty and impact of this work appears to be marginal.
> >### (Q2). Although, there is extensive description on how the current findings are aligned with known results, the authors do not emphasize enough on the new challenges they had to overcome in order to derive their theoretical results or discuss how their work is more challenging from related works.
>
> Answer: We believe that both questions are relevant to the technical novelty of this paper and to technical comparisons with existing works. We refer the reviewer to our response to (Q1) in the ‘response to all reviewers’ part.
>
> >### (Q3). The analysis although insightful is asymptotic in nature which somewhat diminishes the impact of the results.
>
> Answer: Please see our response to (Q2) in the ‘response to all reviewers’ part.
>
> >### (Q4). The structure of the introduction could be improved curving out a related work section.
>
> Answer: We thank the reviewer for the comment regarding the structure of our paper. We acknowledge that in the current structure of our introduction, the discussion of related works is closely intertwined with our motivations for undertaking this research. This integration was intentional, as it helps to contextualize our study within the existing body of literature while simultaneously highlighting the unique aspects and motivations of our work. As such, extracting these discussions into a separate related work section would present some challenges, potentially disrupting the flow and coherence of our narrative.
>
> However, in light of your feedback and the valuable comments from other reviewers, we have taken steps to enhance the 'Influence of Agent’s Sampling Strategy' paragraph in the introduction. This revised part now includes a more detailed technical comparison with existing works. By doing so, we aim to provide a clearer understanding of how our study is positioned relative to the current state of research, while also emphasizing the technical contributions and motivations behind our work. We believe that these revisions will address your concerns about the structure, ensuring that the introduction effectively sets the stage for our research.

---

> > ### Comment · Reviewer_iur4 · 2023-11-23
> > **Post Rebuttal**
> >
> > After thoroughly reading the rebuttal and the comments from the other reviewers I would like to first thank the authors for their efforts to address my concerns. However despite their additional clarifications regarding the theoretical contributions of the paper (although beneficial and helpful in better understanding the new challenges) I still believe that the novelty and impact of these results are limited. On top of that the experimental results presented are insufficient (as noted before) and therefore I am not currently inclined to change my score.

---

### Official Review · Reviewer_bYV1 · 2023-11-08

**Soundness:** 3 good
**Presentation:** 2 fair
**Contribution:** 2 fair
**Rating:** 6
**Confidence:** 3

**Summary:**

This paper provides an asymptotic convergence analysis of generalized distributed SGD (i.e., with a time-varying communication graph, c.f., Kolosokova et al.). It underlines the dependence of the limiting covariance matrix on each client's data-sampling strategy. The paper's main contribution is identifying that while non-asymptotic analyses of GD-SGD using Markovian sampling rely on the mixing time of the worst agent, the asymptotic analysis can benefit from every agent (not just the slowest one), improving their sampling strategies (c.f., Corollary 3.4). Simulations are provided to judge how quickly optimization enters the asymptotic phase and whether client sampling strategies affect the convergence rate.

**Strengths:**

The paper is well-written, and the results are rigorously discussed. The paper highlights an essential difference between asymptotic and finite time bounds and how the latter might sometimes be misleading while looking at client sampling strategies. While the idea of looking at asymptotic regimes and Markovian sampling is not new (as can be seen in Table 1), the paper offers an interesting insight.

**Weaknesses:**

I feel that technical comparison to existing work is lacking. While the table summarizes the existing results and what settings they operate in, it does not discuss what are precisely the bounds obtained by papers such as Doan et al. (2017). As a result, it is unclear whether these bounds actually fail to capture the effect of sampling on all the clients. All the results in this paper require Assumption 2.3. Was that required by the previous papers as well? The experiments at least seem to suggest that an increasing number of local steps is not needed (I am assuming a constant step size was used in the experiments).

**Questions:**

- Can the authors comment on technical comparison to related works, as I mentioned above?
- What were the technical challenges of going to the distributed setting from the known serial analyses? Are any novel techniques needed? Theorem 3.2 seems like a corollary for an existing result.
- What is the step-size schedule in the experiments?

---

> ### Author Response · Authors · 2023-11-19
> **Response to Reviewer bYV1**
>
> We appreciate the reviewer for the detailed comments. Please find our replies to your questions below.
>
> >### (Q1). I feel that technical comparison to existing work is lacking. While the table summarizes the existing results and what settings they operate in, it does not discuss what are precisely the bounds obtained by papers such as Doan et al. (2017). As a result, it is unclear whether these bounds actually fail to capture the effect of sampling on all the clients. All the results in this paper require Assumption 2.3. Was that required by the previous papers as well? Can the authors comment on technical comparison to related works, as I mentioned above?
>
> Answer: When the reviewer mentions [Doan et al. (2017)], we guess it is [Doan et al. (2019)] in our reference. Regarding the technical comparison to existing works, we refer the reviewer to bullet point #2 of the response to (Q1) in the ‘response to all reviewers’ part.
>
> Regarding assumption 2.3, since most of the previous papers focus on the constant communication interval only, we only need assumption 2.3(i), where the usual polynomial step size $γ_n=1/n^a$  for $a∈(0.5,1]$ is assumed. Moreover, the only paper we are aware of talking about the increasing communication interval scheme is [Li et al. (2022)] and our assumption 2.3(ii) is slightly stringent than theirs in order to cover more decentralized settings and Markovian sampling, and we have already included the discussion of this assumption in Remark 1 in our original submission.
>
> >### (Q2). What were the technical challenges of going to the distributed setting from the known serial analyses? Are any novel techniques needed? Theorem 3.2 seems like a corollary for an existing result.
>
> Answer: Please see bullet point #1 of the response to (Q1) in the ‘response to all reviewers’ part.
>
> >### (Q3). What is the step-size schedule in the experiments? The experiments at least seem to suggest that an increasing number of local steps is not needed.
>
> Answer: Thank you for highlighting the omission regarding the step-size schedule in our experiments. Your observation has helped us improve the clarity of our simulation setup. In our experiments, we employ a decreasing step size $γ_n=1/n^{0.9}$. This choice is made to ensure compliance with Assumption 2.3 and thus consistency with the theoretical framework we have established. We have now incorporated this detail into the revised version of our paper.
>
> Regarding the increasing communication interval, our findings depicted in Figure 2(b) reveal that the convergence rate is not adversely affected by this scheme. Contrary to the suggestion that increasing the number of local steps may be unnecessary, our results actually indicate a different conclusion. The implementation of an increasing communication interval scheme effectively reduces the frequency of aggregations, either with neighboring agents or a central server. This strategy leads to a reduction in communication costs, which is a significant consideration in distributed learning scenarios. Importantly, we found that this approach has minimal impact on the convergence speed in the asymptotic regime. This insight is particularly valuable as it suggests that we can achieve cost-effective communication without compromising the efficiency of convergence in distributed learning algorithms.

---

> > ### Comment · Reviewer_bYV1 · 2023-11-23
> >
> > I thank the authors for their detailed response. I have gone through the responses and the other reviews. I have decided to retain my score. I believe there are additional technical challenges over previous works, such as dealing with consensus error and non-iid sampling. But again, any extension of serial results to the distributed setting must deal with that analysis. So overall, my impression is that the work closely builds on existing tools in the literature. I appreciate the asymptotic viewpoint and concede that the algorithms presented hit the asymptotic regime in the experiments, thus making it worthwhile to study the regime.
> >
> > Regarding the experiments, the current experiments do validate the theoretical results, when the local steps are not growing, using $a=0.9$. However, why do the authors use this step size when logarithmically growing the local steps? Overlooking this issue, I would have liked to see how other step-size schemes pan out. In particular, if the step size were tuned optimally (i.e., tuning $a$) for each instance for a fixed number of time steps, I would imagine shifting the asymptotic regime for different instances. This could, in turn, change the relative performance of different sampling schemes. The non-convex experiments offered in the appendix are very "convex-like", and it would be good to have more comprehensive experiments using even a simple neural network. Finally, the simulation doesn't have data heterogeneity. I believe this makes it harder to comprehend the differences between the agents, which the paper claims is a benefit of the asymptotic analysis over the non-asymptotic one.
> >
> > The authors write in response to the reviewer qxNb:
> >
> > > The core of our contribution lies in the generality of our model and the unified results we offer within the broad framework of decentralized learning, where we uncover that the sampling strategy of each individual agent affects the overall performance while the effect of communication pattern contributes only via its leading eigenvector (stationary distribution) under the most general setup.
> >
> > In the current write-up, this takeaway is obfuscated by the technical results. The discussion below corollary 3.4 can be revised with more examples. This relates to the limitation that the authors do not discuss the practical relevance of their results. Which federated learning applications can benefit from non-iid sampling, or where can the Markovian sampling suggested in this paper be implemented efficiently? Is there a natural decentralized setting where this is possible? What is the additional computational cost of doing this? How can this be implemented in online settings where the data is not stored on the device? Some of these questions might have simple answers, but providing this context is important, otherwise, the work comes off as a mechanical composition of two existing techniques: asymptotic analyses and consensus error-based analyses---something most reviewers have complained about. While there is not much time left in the discussion period, hopefully the authors can address these issues in their revision.

---

### Author Response · Authors · 2023-11-19
**Response to All Reviewers (1/2)**

We thank all the reviewers for their comments and for the time and effort they put into reading, understanding, and evaluating our paper. Their insightful comments have been pivotal in improving our paper. We have noticed that several concerns raised by the reviewers overlap. To address these efficiently and avoid repetition, we are providing a collective response below. Each point of concern has been carefully considered, and we have outlined our responses and the corresponding changes (in blue) to the revised version.

>### (Q1). The technical contribution of the paper is insufficient, and can be seen as a straightforward extension to the previous works.

Answer: To respond to this question, we consider the following two aspects:

### 1. **Technical novelty compared to existing works**:

Reviewers qXnB and iur4 raised the concern that our technical results are straightforward extension to the works of [Li et al. (2022), Hu et al. (2022), Morral et al., (2017), Koloskova et al., (2020)]. However, our proof methodology and analysis significantly diverge from those in the mentioned studies.

While [Li et al. (2022)] (resp. [Morral et al., (2017)]) focus on the CLT analysis in a local SGD setting with increasing communication interval (resp. in a decentralized SGD setting with a constant (fixed) communication interval $K=1$) with i.i.d. sampling and a specific communication pattern $\\mathbf{W}\_n=\\mathbf{1}\\mathbf{p}^T$ (resp. a deterministic doubly stochastic communication matrix $\mathbf{W}$) for some probability vector $\mathbf{p}$, our work on GD-SGD encompasses a broader range of communication patterns and a Markovian sampling strategy for each agent. This complex framework is a notable departure from the aggregation model of [Li et al. (2022), Morral et al., (2017)] and represents a more generalized and intricate system. Additionally, while [Koloskova et al., (2020)] consider various communication patterns, it only assumed i.i.d. sampling for each agent, ruling out the possibility of employing Markovian sampling for each agent.

Unlike [Hu et al. (2022)], which examine the effects of sampling strategies in SGD for a single agent by leveraging the existing CLT result, our study extends this analysis to a multi-agent context under the GD-SGD framework. This extension is far from trivial; it requires addressing unique challenges such as the consensus error among agents under *varying communication patterns* and *increasing communication interval*, as detailed in Lemma C.1 in our Appendix C. We successfully demonstrate the pathwise boundedness of a sequence $\\{\\phi\_n\\}$, representing the consensus error scaled by the inverse of the step size. This finding, leading to asymptotically zero consensus error almost surely, is a novel contribution not found in [Li et al. (2022), Hu et al. (2022), Morral et al., (2017), Koloskova et al., (2020)].

Furthermore, in our approach, the noise term $\\nabla F_i (\\theta^i,X^i )-\\nabla f_i (\\theta^i)$ for each agent $i$ is not necessarily unbiased due to the Markovian sampling strategy, posing significant technical difficulties compared with i.i.d. sampling considered in [Li et al. (2022), Morral et al., (2017), Koloskova et al., (2020)]. While [Hu et al. (2022)] mention the closed-form solution of the Poisson equation, we extend this technique by decomposing the Markovian noise term $\\nabla F_i (\\theta^i,X^i )-\\nabla f_i (\\theta^i)$ for each agent into several terms in (57) and thoroughly quantify each term's properties in our original Appendices D.1, D.2 (for the almost sure convergence), and E.1 to E.3 (for the CLT). This detailed decomposition and quantification of noise terms in a multi-agent context are also absent in the previous works, e.g., [Li et al. (2022), Hu et al. (2022); Morral et al., (2017); Koloskova et al., (2020)].

To accommodate reviewer’s comment, we have revised the paragraphs after Lemma 3.1, Theorem 3.2 and Theorem 3.3 to include more insights of our analysis towards our main results.

---

> ### Author Response · Authors · 2023-11-19
> **Answer to (Q1) Cont.**
>
> ### 2. **The limitation of current finite-time analysis**:
>
> Finite-time analyses, while valuable, fail to capture the nuanced impact of each individual agent’s sampling strategy to the same extent as our asymptotic approach. The finite-time bound obtained in [Doan et al. (2019)] is of the form
> $$\\mathbb{E}[‖θ_n^i-θ^* ‖^2 ]≤O(\\log⁡(n+1)/(n+1))$$
> for step size $γ_n=1/(n+1)$ and any agent i, where the big O term does not include any agent-specific constant. [Zeng et al. (2022)] explicitly define the mixing time of the worst-performing agent in (5) therein and extend the bound of [Doan et al. (2019)] to more general objective function with an additional factor $\\log⁡(n+1)$.
>
> Meanwhile, the finite-time bound in [Wai (2020), Sun et al. (2023)] is of the form
> $$\\min_{1≤k≤n}\\mathbb{⁡E}[‖∇f(\bar{θ}_k-θ^* )‖^2 ]=O\\left(\\frac{1/\\log^2⁡(1/ρ)}{n^{1-a}}\\right)$$
> where $\\bar\\theta_k$ is some weighted average over all agents, $ρ$ is the mixing rate of the underlying Markov chain of each agent, which is assumed to be identical for all agents. Therefore, the existing finite-time bounds are unable to account for agent-specific influences, either assuming identical sampling strategy across agents or basing analyses on the worst-performing agent with largest mixing rate $ρ$. In contrast, our asymptotic approach provides a more detailed and accurate representation of the impact of each agent’s sampling strategy on the overall system performance.
>
> To better address the reviewer’s concerns, we have revised the paragraph `Influence of Agent’s Sampling Strategy’ in the introduction to show the finite-time bounds of existing works and demonstrate how these founds fail to capture the effect of sampling strategy of each agent.
>
> >Morral, G., Bianchi, P., & Fort, G. Success and failure of adaptation-diffusion algorithms with decaying step size in multiagent networks. IEEE Transactions on Signal Processing, 65(11), 2798-2813, 2017.
> >
> >Koloskova, A., Loizou, N., Boreiri, S., Jaggi, M., & Stich, S. A unified theory of decentralized sgd with changing topology and local updates. In International Conference on Machine Learning (pp. 5381-5393). PMLR, 2020.
> >
> >Li, X., Liang, J., Chang, X., & Zhang, Z. Statistical estimation and online inference via local sgd. In Conference on Learning Theory (pp. 1613-1661). PMLR, 2022.
> >
> >Hu, J., Doshi, V., & Eun, D. Y. Efficiency Ordering of Stochastic Gradient Descent. Advances in Neural Information Processing Systems, 35, 15875-15888, 2022.

---

### Author Response · Authors · 2023-11-19
**Response to All Reviewers (2/2)**

>### (Q2). The results established in this paper are in an asymptotic sense which somewhat diminishes the impact of the results; can these results be extended to non-asymptotic ones?

Answer: In our work, we specifically focus on the asymptotic analysis to showcase the impact of individual agents' sampling strategies on the overall performance of the GD-SGD algorithm. As we have demonstrated in the paragraph `Rationale for Asymptotic Analysis’, while recent trends have shown a preference for non-asymptotic analysis, CLT is far less asymptotic than it may appear for SGD under both i.i.d. and Markovian sampling [Mou et al.(2020); Chen et al.(2020)]. In addition, discussions in [Meyn (2022), Chapter 1.2] point out the often-underestimated significance of asymptotic statistics.

Since we have mentioned in the bullet point #1 of the previous answer that one of our technical novelties lies in the decomposition of the GD-SGD iterates into a stochastic-approximation-like form in (57) in Appendix D, it is possible to apply the non-asymptotic analysis in the stochastic approximation literature to derive the finite sample bound. While this formulation opens up the possibility of applying non-asymptotic analysis methods from the stochastic approximation literature to derive finite sample bounds, we chose not to pursue this direction in our current work.

The rationale behind this decision stems from the specific focus of our study on the impact of individual agents' sampling strategies on the overall performance of the GD-SGD algorithm. Current non-asymptotic analysis techniques do not adequately differentiate the contributions of individual agents within the decentralized learning framework, which has been discussed in the bullet point #2 of our previous answer. Furthermore, even an improved non-asymptotic analysis that differentiates mixing rates among agents would not fully represent the true performance dynamics of the underlying Markov chains. As noted in [Hu et al. (2022)], relying solely on mixing rates for qualitative comparisons between Markov chains in optimization algorithms can be misleading. In contrast, the asymptotic covariance matrix, as derived from our CLT results, provides a more detailed and meaningful evaluation of various sampling strategies.

>Mou, W., Li, C. J., Wainwright, M. J., Bartlett, P. L., & Jordan, M. I. On linear stochastic approximation: Fine-grained Polyak-Ruppert and non-asymptotic concentration. Conference on Learning Theory. PMLR, 2020.
>
>Chen, S., Devraj, A., Busic, A., & Meyn, S. Explicit mean-square error bounds for monte-carlo and linear stochastic approximation. In International Conference on Artificial Intelligence and Statistics (pp. 4173-4183). PMLR, 2020.
>
>Meyn, S. Control systems and reinforcement learning. Cambridge University Press, 2022.

---

### Author Response · Authors · 2023-11-21
**Reminder to All Reviewers**

Dear Reviewers,

We would like to remind you that the rebuttal period will close in one day. During this phase, we have responded to the reviews and comments/concerns you have provided. Your engagement is crucial in this part of the process. Your insights and expertise are invaluable to ensuring the improvement of our paper. We greatly appreciate your commitment and time dedicated to this important phase.

Thank you once again for your valuable reviews.

Best regards,

Submission #412 Authors

---

### Meta-Review · Area_Chair_Zjxh · 2023-12-06

**Metareview:**

This paper offers an in-depth evaluation of the asymptotic convergence analysis of generalized distributed (decentralized) Stochastic Gradient Descent (GD-SGD) focusing on varying communication patterns and sampling strategies,  including iid sampling and Markovian sampling.  The influence of the sampling strategies on the limiting covariance matrix is analyzed.
One of the main contributions is identifying that while non-asymptotic analyses of GD-SGD using Markovian sampling rely on the mixing time of the worst agent, the asymptotic analysis can benefit from every agent (not just the slowest one).

The the reviewers found the paper to be well written and clear, however, two aspects were found to be a limitation by some of the reviewers:
- the asymptotic analysis (vs. non-asymptotic). The authors argued convincingly that so far prior non-asymptotic analysis could not yet capture the impact of each individual agent’s sampling strategy in such a nuanced as done in this work.
- the experimental results are confined to the CIFAR10 dataset, calling for a wider experimental scope.

The reviewers opinion's on this paper were divided. On the one hand, the paper provides novel insights. On the other hand, the constrain to asymptotic analysis may hinder the broad applicability and generalizability of the findings.

**Justification For Why Not Higher Score:**

The perceived lack of novelty by some reviewers (the technical challenges were still a bit vague, even after the rebuttal), restriction to asymptotic paradigms, and limited comparison with existing work indicate areas for improvement.

**Justification For Why Not Lower Score:**

N/A

---

### Decision · Program_Chairs · 2024-01-16

Reject